# Green Synthesized Polymeric Iodophors with Thyme as Antimicrobial Agents

**DOI:** 10.3390/ijms25021133

**Published:** 2024-01-17

**Authors:** Zehra Edis, Samir Haj Bloukh, Hamed Abu Sara, Iman Haj Bloukh

**Affiliations:** 1Department of Pharmaceutical Sciences, College of Pharmacy and Health Science, Ajman University, Ajman P.O. Box 346, United Arab Emirates; 2Center of Medical and Bio-Allied Health Sciences Research, Ajman University, Ajman P.O. Box 346, United Arab Emirates; s.bloukh@ajman.ac.ae (S.H.B.); h.abusara@ajman.ac.ae (H.A.S.); 3Department of Clinical Sciences, College of Pharmacy and Health Science, Ajman University, Ajman P.O. Box 346, United Arab Emirates; 4College of Dentistry, Ajman University, Ajman P.O. Box 346, United Arab Emirates; imansamir835@gmail.com

**Keywords:** green synthesis, antimicrobial resistance, thyme, *Aloe Vera*, smart triiodides, surgical-site infection, face mask, gauze bandage, iodophors, sustainability

## Abstract

Antimicrobial resistance (AMR) is a growing concern for the future of mankind. Common antibiotics fail in the treatment of microbial infections at an alarming rate. Morbidity and mortality rates increase, especially among immune-compromised populations. Medicinal plants and their essential oils, as well as iodine could be potential solutions against resistant pathogens. These natural antimicrobials abate microbial proliferation, especially in synergistic combinations. We performed a simple, one-pot synthesis to prepare our formulation with polyvinylpyrrolidone (PVP)-complexed iodine (I_2_), *Thymus Vulgaris* L. (Thyme), and *Aloe Barbadensis* Miller (AV). SEM/EDS, UV-vis, Raman, FTIR, and XRD analyses verified the purity, composition, and morphology of AV-PVP-Thyme-I_2_. We investigated the inhibitory action of the bio-formulation AV-PVP-Thyme-I_2_ against 10 selected reference pathogens on impregnated sterile discs, surgical sutures, cotton gauze bandages, surgical face masks, and KN95 masks. The antimicrobial properties of AV-PVP-Thyme-I_2_ were studied by disc diffusion methods and compared with those of the antibiotics gentamycin and nystatin. The results confirm AV-PVP-Thyme-I_2_ as a strong antifungal and antibacterial agent against the majority of the tested microorganisms with excellent results on cotton bandages and face masks. After storing AV-PVP-Thyme-I_2_ for 18 months, the inhibitory action was augmented compared to the fresh formulation. Consequently, we suggest AV-PVP-Thyme-I_2_ as an antimicrobial agent against wound infections and a spray-on contact killing agent.

## 1. Introduction

The recent COVID-19 pandemic left mankind traumatized and deeply wounded. Deep insecurity caused by shortages of medication, personal protective equipment (PPT), and overwhelmed medical facilities marked the disaster [1,2,3,4]. The uncontrollable, rapid transmission of the virus by air droplets caused mounting hospitalization and fatality rates globally [1,2,3,4]. COVID-19 enhanced the already ongoing problem of antimicrobial resistance (AMR) by an overuse of antimicrobials to save severely ill, co-morbid patients in emergency rooms [1,2,3,4,5,6]. Evolving multi-drug-resistant ESKAPE pathogens (*Enterococcus faecium*, *Staphylococcus aureus*, *Klebsiella pneumoniae*, *Acinetobacter baumannii*, *Pseudomonas aeruginosa*, *Enterobacter* spp., and *Escherichia coli*) are already reasons for exacerbating treatment duration, costs, and concern for several years [5,6,7,8,9,10,11,12,13,14,15]. ESKAPE pathogens linger in emergency units and hospital wards, where they escalate morbidity and mortality rates [5,6,7,8,9,10,11,12,13,14,15]. Investigators gain constantly more insight into the microbial morphology, properties, structure, and virulence factors that enable better defense strategies [8,9,10,11,12,13,14].

Medicinal plant-derived new generations of natural antimicrobial agents could support or replace ineffective common drugs [16,17,18,19,20,21,22]. Plants incorporate a plethora of bioactive compounds, which synergistically defend them against opportunistic pathogens and further threats since their existence [21,22]. The synergistic use of plant bioactive compounds could be an alternative to mitigate AMR-related outcomes. The increasing number of investigations confirms the endless opportunities derived from the plant kingdom. Different medicinal plants in one formulation exacerbate antimicrobial properties and form a new defense line against AMR. A mounting number of studies report successful combinations of plant extracts with different nanoparticles (NP). Nanomedicine can be an option to improve health outcomes in the future through better drug delivery, precision medicine, and biomedical applications [23]. Our group investigated iodine-based formulations, consisting of silver nanoparticles (AgNP) and/or plant extracts with inhibitory action against selections of reference microorganisms [24,25,26,27,28,29,30,31,32,33,34,35,36]. AMR remains a serious threat to the future of mankind because opportunistic pathogens develop resistance against many synthetic antimicrobials on the market. Iodine is a strong microbicide and has until now not been associated with any resistance phenomenon, although it has been used since history. Therefore, iodine-based formulations have the potential to be used with the purpose of inhibiting microorganisms. Adding iodine to the formulation enhances its antifungal and antibacterial properties, in agreement with our previous works. However, iodine must be protected by PVP in order to have long-lasting action. Here, the PVP-I_2_ complex acts as a slow-release reservoir of iodine molecules. Additionally, synergy between plant biocomponents and the PVP-I complex enriches the inhibition portfolio.

The materials we used in our group included *Aloe Barbadensis* Miller (Aloe Vera, AV), *Salvia officinalis* L. (Sage), *Cinnamomum zeylanicum* (Cinn), trans-cinnamic acid (TCA), iodine, polyvinylpyrrolidone (PVP), and AgNP [31,32,33,34,35,36].

*Aloe Barbadensis* Miller (Aloe Vera, AV) is established as a remedy for ailments in different cultures because of its diverse antimicrobial, anti-inflammatory, and soothing properties [37,38,39,40,41,42,43,44,45,46,47,48,49,50,51,52,53,54,55,56,57,58,59,60,61,62,63,64,65,66]. AV contains more than 75 bio-compounds, which include, for example, acemannan, aloin, emodin, galacturonic acid, and mannose [23,34,35,36,56,57,58,59,60,61,62,63,64,65,66]. Therefore, AV products are exponentially growing due to increasingly health-conscious consumers [42]. AV is utilized in oral care and hygiene products to prevent or mitigate inflammation, oral ulcers, plaque formation, gingivitis, and periodontal disease [49,50,51,52,67,68]. Biofilm formation on tooth surface areas is a major problem that causes dental caries and gingivitis over time [51,52,67,68,69]. Mouthwash, toothpastes, and hydrogels for wound treatment are common applications [49,50,51,52]. We utilized AV gel due to its antimicrobial, moisturizing, and health-promoting properties. Additionally, AV is a globally available, cost-effective, and sustainable resource [23,34,35,36,56,57,58,59,60,61,62,63,64,65,66].

AV and further medicinal plants, including their essential oils, have known antimicrobial, anti-inflammatory, and antioxidative properties [5,70,71,72,73]. Most of them are low-cost, easily available ingredients that can be found in almost every household. Such medicinal plants and their essential oils could play a pivotal role against AMR due to their antimicrobial properties and are therefore increasingly utilized in fields like wound and oral care [5,73,74,75,76]. *Thymus Vulgaris* L. (Thyme) grows in many parts of the world and has been known since history as a health-promoting plant [23,70,71,72,73,76,77,78,79]. The main components of Thyme extract are the monoterpenes thymol and carvacrol, which gained popularity as antimicrobial agents [73,80,81,82,83,84,85,86,87,88]. Depending on the type of extraction and the plant material, further components include phenolic acids like rosmarinic acid, gallic acid, caffeic acid, vanillic acid, syringic acid, and chlorogenic acid, as well as the flavonoids luteolin and apigenin [70,71,72,73,74,75,76,77,78,79,80,81,82,83,84,85,86,87,88,89,90,91,92,93,94]. These compounds are known for their antimicrobial properties and are involved, for example, in pharmaceutical, cosmeceutical, medicinal, and dental applications [5,16,17,18,34,36,70,71,72,73,74,75,76,77,78,79,80,81,82,83,84,85,86,87,88,89,90,91,92,93,94]. Some of those include antimicrobial food packaging, oral and skin care products, wound, plaque- and biofilm-preventing treatments, as well as personal protective equipment (PPE) [5,16,17,18,34,36,70,71,72,73,74,75,76,77,78,79,80,81,82,83,84,85,86,87,88,89,90,91,92,93,94,95,96,97,98,99,100,101,102]. Face masks are part of the PPE imperative in mitigating the proliferation of opportunistic microorganisms [2,4]. They form a barrier to microorganisms by reducing or preventing their access to the human respiratory tract and oral and nasal cavities [34,36,95,96,97,98,99,100,101]. We previously investigated formulations of AV gel with *Salvia officinalis* L. (Sage) and trans-cinnamic acid (TCA) in combination with polyvinylpyrrolidone (PVP) and iodine [34,35,36]. Our formulations (AV-PVP-I_2_, AV-PVP-Sage-I_2_, and AV-PVP-TCA-I_2_) showed interesting inhibitory action when impregnated on sterile discs, surgical sutures, cotton bandages, and surgical face masks [34,35,36]. Incorporating iodine into the formulations amplified the antimicrobial properties of our formulations and resulted in enlarged inhibitory zones on face masks, discs, sutures, and cotton bandages [34,35,36].

Iodine (I_2_) is known and utilized as a microbicide historically but is entailed with some side effects, including skin irritation, pain, skin discoloration, and uncontrolled iodine release [29,30,34,35,36,103,104,105,106,107,108]. Iodine forms different moieties consisting of molecular iodine, iodide ions, and tri and higher polyiodide anions, depending on the surrounding molecular structure [24,25,26,27,28,29,30,32,34,35,36,109,110,111,112,113]. Triiodides are the most stable form of polyiodides [29,30,34,35,36,109,110,111]. They have a negative charge and can form, according to local molecular surroundings, the inherently stable, symmetrical, linear “smart” units [I-I-I^−^] or the less stable, nonlinear [I-I⋯I^−^] moieties [29,30,34,35,36,109,110,111]. Nonlinear [I-I⋯I^−^] moieties are more prone to uncontrolled iodine release and, therefore, abate the long-term stability and treatment durations [29,30,34,35,36,109,110,111]. The iodine release mechanisms can be controlled by ensuring the formation of “smart” triiodides [I-I-I^−^] through the defined surroundings. The same effects can be achieved by simply stabilizing polyiodide groups within a matching complexing agent [29,30,34,35,36,109,110,111]. Polymeric matrices like polyvinylpyrrolidone (PVP) are used in many products in combination with iodine in the form of PVP-I_2_ and act as iodophor [29,34,35,36,103,104,105,106,114,115,116,117,118,119,120,121,122,123,124,125,126,127,128,129]. PVP-I_2_, also called povidone iodine, is a basic microbicide and is available globally [29,34,35,36,103,104,105,106,125,126,127,128,129]. It is widely utilized as a gargle or mouthwash. It can also be used during the treatment of wounds and in surgical procedures for the prevention of infections and support of the healing process [34,35,36,103,104,105,106,107,108,125,126,127,128,129].

This study is based on our previous investigations related to “smart” triiodides and AV biohybrids [34,35,36]. Our new plant-based formulation AV-PVP-Thyme-I_2_ consists of Thyme maceration extract in synergy with AV gel, PVP, and iodine in accordance with our previous studies [34,35,36]. The bio-formulation AV-PVP-Thyme-I_2_ was studied using the analytical methods such as SEM/EDS, Raman spectroscopy, UV-vis, FTIR, and X-ray diffraction (XRD), which confirmed its purity and morphology. Our aim is to offer facile, non-toxic, sustainable, low-cost antimicrobial agents prepared by readily available basic ingredients (Thyme, AV, and PVP-I_2_) through one-pot synthesis. Such formulations can be potential solutions against AMR and could improve quality of life in times of crisis with supply shortages related to disinfectants, PPE, or antimicrobial agents.

Antimicrobial properties of the title bio-formulation AV-PVP-Thyme-I_2_ were verified using disc diffusion methods against 10 of our reference strains in comparison to gentamycin and nystatin. The studied microorganisms consisted of the fungus *C. albicans* WDCM 00054 Vitroids; the Gram-positive bacteria *S. pneumonia* ATCC 49619, *S. aureus* ATCC 25923, *S. pyogenes* ATCC 19615, *E. faecalis* ATCC 29212, and *B. subtilis* WDCM0003; as well as the Gram-negative *E. coli* WDCM 00013 Vitroids, *P. mirabilis ATCC 29906*, *P. aeruginosa* WDCM 00026 Vitroids, and *K. pneumonia* WDCM00097 Vitroids. Storing AV-PVP-Thyme-I_2_ for 18 months in the fridge in darkness ameliorated the antimicrobial properties against the same reference strains. We also investigated the inhibitory action of the biohybrid AV-PVP-Thyme-I_2_ by impregnating surgical PGA sutures, cotton bandages, and disposable surgical facemasks and KN95 facemasks.

*C. albicans* WDCM 00054 Vitroids was highly susceptible to AV-PVP-Thyme-I_2_, followed by Gram-positive and Gram-negative pathogens. The KN95 mask, the surgical face mask layers, and the cotton bandages showed the highest zones of inhibition (ZOI) against the microorganisms in descending order. The title formulation shows enhanced antimicrobial properties even after 18 months of storage. These results suggest the use of the bio-hybrid as an antifungal and antibacterial agent on KN95, surgical face masks, and cotton bandages to mitigate infections. They can be potentially utilized as spray-on contact killing agents on those materials, oral cavities, wounds, and inanimate surfaces in public or hospital settings. Future investigations are needed to verify the suggested applications of our bio-formulation AV-PVP-Thyme-I_2_ in the form of cell culture and cytotoxicity studies, as well as in vivo experiments.

## 2. Results and Discussion

AMR is marked by increased strains of multi-drug-resistant pathogens that are not susceptible to common antimicrobials [1,2,3,4,130]. Such treatment failures prove to be fatal, especially among elderly and immune-compromised patients affected by nosocomial infections [5,6,7,8,9,10,11,12,13,14,15,130]. ESKAPE pathogens are an inherent danger for the future of mankind [2,3,4,130]. Opportunistic pathogens like *S. aureus*, *E. faecalis*, *S. pyogenes*, *E. coli*, *P. aeruginosa*, and *C. albicans* can colonize oral cavities and form biofilms on and around teeth, as well as on the dentin walls of root canals [131,132,133]. *C. albicans* and *E. faecalis* colonization can lead to dental biofilms, inflammation, plaque formation, dental caries, periodontal disease, and other dangerous systemic infections by entering the blood, like rheumatoid arthritis, chronic kidney disease, and inflammatory bowel disease [131,132,133]. *S. pyogenes* isolated from dental plaque has the ability to result in throat and oral cavity infections, which can progress towards life-threatening illnesses like pneumonia, endocarditis, or encephalitis [133].

Plant-based bio-antimicrobials can be an alternative to failed classes of synthetic antimicrobials [16,17,18,19,20,21,22,23,34,35,36,70,71,72,73]. Wound and surgical-site infections can be potentially prevented through synergistic combinations of plant bio-compounds [16,17,18,19,20,21,22,23,34,35,36,39,40,41,42,43,44,45,46,47,48,69,70,71,72,73]. Adding iodine protected within the polymeric matrix PVP, a sustained-release reservoir, enhances inhibitory action towards opportunistic pathogens [34,35,36,125,126,127,128,129]. Such combinations can potentially reduce chronic inflammatory processes as well as pain, suffering, treatment costs, and duration [34,35,36,125,126,127,128,129]. Therefore, we investigated the morphology, composition, and antimicrobial properties of the title formulation AV-PVP-Thyme-I_2_. The inhibitory action of fresh and 18 months stored bio-material AV-PVP-Thyme-I_2_ was tested against a selection of 10 reference strains on sterile discs, surgical PGA sutures, cotton gauze bandages, surgical face masks, and KN95 masks.

### 2.1. Elemental Composition and Morphological Examination of AV-PVP-Thyme-I_2_

Scanning electron microscopy (SEM) and energy-dispersive X-ray spectroscopic (EDS) analyses were performed to analyze the morphology and composition of AV-PVP-Thyme-I_2_, respectively (Figure 1).

AV-PVP-Thyme-I_2_ shows an interesting morphology consisting of amorphous round-shaped entities with a smooth surface and semi-crystalline, flaked structures covered with white cube-like or almost spherical spots or lines (Figure 1a and Appendix A). The EDS reveals carbon (40%), oxygen (39%), and iodine (4%) (Figure 1b). Aluminum with 13% is a result of the sample preparation for the SEM, while chlorine (1.7%), potassium (1.6%), sodium (0.4%), and iron (0.3%) are from the thyme extract and the AV gel [90]. Au appears as well in the EDS because of the gold coating.

The formulation AV-PVP-Thyme-I_2_ (11 µg/mL) was impregnated on sterile surgical face masks, bandages, and braided polyglycolic acid (PGA) sutures to investigate the inhibitory action against a selection of 10 microorganisms. Figure 2 represents the SEM and EDS of dip-coated surgical PGA sutures.

Figure 2b and c shows the SEM of the braided surgical PGA suture coated with AV-PVP-Thyme-I_2_. These can be compared to our previous investigation showing the same, uncoated, plain PGA suture (Figure 2a) [34]. The coated suture has clear patches of semi-crystalline nature distributed all over. These patches are covering the braids of the suture homogenously. The EDS reveals again carbon (53%), oxygen (40%), iodine (2%), and chlorine (0.2%) (Figure 2d and Appendix A). Cupper with 4% originates from the suture itself. The homogenously coated PGA sutures have a potential to be used in surgical operations to prevent surgical-site infections. Figure 3 depicts the SEM analysis of impregnated surgical facemasks (Appendix A).

The medical face masks consisted of two layers. The white inner layer, which is directed towards the face, and the blue outer layer. The morphology of the two layers under the microscope differs from each other completely. The white layer shows a dense structure of entangled thin fibers (Figure 3a), while the blue outer layer shows a net-like structure with robustly ordered fibers next to each other (Figure 3c). Figure 3b,d shows semi-crystalline depositions of small, cube-like or spherical moieties on the surface of the face mask tissues. Figure 3c,d reveals a thin layer of coating on the tissue surface, while Figure 3b shows hives of title bio-material on the originally white, thin, entangled fibers. The thin, homogenous layer on the blue face mask tissues indicates successful adsorption. As a conclusion, the formulation AV-PVP-Thyme-I_2_ is homogenously distributed on the face mask tissues and does not compromise breathability or air movement (Figure 3a,b). Therefore, our impregnated face mask tissues could be used to mitigate viral load and inflammatory processes in the upper respiratory tract. However, further studies are needed to confirm the antiviral activity of our formulation, which will be part of our future investigations.

SEM and EDS analyses were performed on sterile bandages coated with the formulation AV-PVP-Thyme-I_2_ (Figure 4 and Appendix A).

The cotton bandage is homogenously coated by the title biohybrid (Figure 4a,b). In Figure 4b, the same pattern of small-sized, almost spherical or cube-like patches is seen as previously confirmed in the face mask tissue in Figure 3d. Additionally, the whole surface is uniformly coated by these small circular or cube-like patches. Figure 4c depicts the EDS of the title formulation on the bandage. Carbon with 65.1% is followed by oxygen with 31.8, and finally iodine with 3.1%. Again, gold appears as a result of the gold coating. The homogenous deposition of AV-PVP-Thyme-I_2_ on the bandage is a good indicator for the potential use of our impregnated bandages as antimicrobial dressings.

### 2.2. Spectroscopical Characterization

#### 2.2.1. Raman Spectroscopy

Figure 5 is the result of the Raman spectroscopic analysis of the title bio-material AV-PVP-Thyme-I_2_.

The Raman analysis of the title formulation AV-PVP-Thyme-I_2_ in Figure 5 confirms the presence of polyiodide moieties in accordance with our previous studies [29,30,34,35]. The adsorption band at 112 cm^−1^ indicates triiodide ions, which consist of purely symmetrical I_3_^−^-units (Figure 5 and Table 1) [29,30,34,35,36,81,111,118,129].

The reason behind the absorption peak at 112 cm^−1^ are the symmetrical vibrations ν_1s_ caused by symmetrical, “smart” triiodide ions (I-I-I^−^), which usually are seen around 100–115 cm^−1^ (Figure 5 and Table 1) [29,30,34,35,36].

The weak, broad shifts at 141 and 145 cm^−1^ belong to unsymmetrical triiodide units made up of molecular I_2_ and iodide ions in the form of (I-I⋯I^−^) as overtones (Figure 5 and Table 1) [29,34,35,36,111]. These are distorted, unsymmetrical, and nonlinear triiodide units (I-I⋯I^−^) and are also confirmed by UV-spectral analysis in the next part of this investigation (Figure 5 and Table 1) [29,34,35,36]. The Raman spectrum does not indicate the presence of any other iodine moieties. The absence of strong absorption bands between 140 and 175 cm^−1^, as well as the missing absorption bands around 166 cm^−1^ confirms definitely the lack of pentaiodide ions within the title bio-material (Figure 5 and Table 1) [29,34,35,36]. The absence of other Raman shifts confirms the purity of AV-PVP-Thyme-I_2_.

#### 2.2.2. UV-Vis Spectroscopy

Figure 6 depicts the UV-vis spectra of fresh AV-PVP-Thyme-I_2_ in comparison to AV-PVP-Thyme-I_2_ after 18 months of storage, AV-PVP-Thyme, PVP-I_2_, and pure Thyme macerated for 3 months.

The UV-vis spectral analysis reveals interesting information when the samples AV-PVP-Thyme, AV-PVP-Thyme-I_2_, AV-PVP-Thyme-I_2_ (after 18 months), pure Thyme macerated for 3 months, and PVP-I_2_ are compared (Figure 6a–c). As witnessed in our previous investigations, similar patterns of high-intensity, broad absorptions appear around 200–240, 250–320, and 325 to 420 nm in the iodinated samples (Figure 6) [29,34,35,36]. The absorptions of AV- and Thyme-bio-compounds, PVP, and iodine units overlap with each other and cause broad bands [29,34,35,36]. Thyme and AV-bio-compounds are clearly over-dominated by the stronger intensity absorptions of PVP and iodine moieties, which makes the spectral analysis challenging [29,34,35,36]. Therefore, additional UV spectra of pure Thyme extract, which was macerated in ethanol for 3 months, and its main bio-constituents thymol, and carvacrol were added to the investigation. The comparison with these and the previously studied AV-PVP-Sage-I_2_ allows better predictions on the composition of the title formulation since Sage contains thymol and carvacrol (Table 2) [34].

The formulation AV-PVP-Thyme-I_2_ shows in the UV-visible spectrum strong absorption bands related to I_2_ at 204 nm, I^−^ at 201 nm, and “smart” triiodide ions in the form of symmetrical, linear [I-I-I^−^] units at 289 and nonlinear [I-I⋯I^−^] moieties at 359 nm, in agreement with our previous studies (Figure 6 and Table 2) [24,25,26,27,28,29,30,32,33,34,35,36]. After storing the AV-PVP-Thyme-I_2_ for 18 months (blue curve), the UV-vis spectrum reveals almost similar absorptions without change. It is suggested that the triiodide absorption intensities, as well as their increased band broadness, indicate not only enhanced hydrogen bonding within the formulation but also a predominance of symmetrical [I-I-I^−^] in comparison to asymmetrical triiodide [I-I⋯I^−^] moieties. This assumption is supported by the Raman spectrum, which shows a strong absorption peak at 112 cm^−1^ for “smart” triiodides and a broad, weak absorption band at 141 and 145 cm^−1^ for asymmetric [I-I⋯I^−^] units (Figure 5 and Table 1) [29,30,34,35,36]. The absence of pentaiodide ions in the title biohybrid is not only confirmed by the Raman spectrum but also in the UV-spectral analysis. The purple curve for PVP-I_2_ reveals a broad band around 444 nm as an indicator for I_5_^−^ units, while it is lacking in AV-PVP-Thyme-I_2_ (red curve) and AV-PVP-Thyme-I_2_ (after 18 months, blue curve) (Figure 6 and Table 2).

Comparing the available UV spectra allows predictions about possible interactions within the different samples (Figure 6 and Table 2). The UV analysis of pure Thyme extract (orange curve), which was macerated for three months, reveals four broad absorption bands from 235 to 260, 265 to 290, 310 to 360, and 380 to 430 nm, ordered in descending absorbance intensity from weak to very weak (Figure 6c). These absorptions are a result of the bio-compounds within the Thyme extract, which are the known major components thymol and carvacrol, among several others [76,77,78,79,80,81,82,83,84,85,86,87,88,89,90,91,92,93,94]. Once AV-PVP is added (AV-PVP-Thyme, green curve), the intensities decrease vehemently due to a hypsochromic effect as a result of the higher number of hydrogen bonds with PVP. During this process, biocomponents within Thyme are encapsulated by PVP, leading to a decrease in their conjugated systems and chromophores (Figure 6 and Table 2).

Adding iodine increases the absorption of triiodide moieties highly (Figure 6, red curve). This curtails the follow-up of the Thyme- and AV-biocomponents whereabouts through overlap with the strong, broad, high-intensity bands of the triiodide units around 289 and 359 nm in the red curve. Further, increased absorption around 289 and 359 nm occurs after storing AV-PVP-Thyme-I_2_ for 18 months (blue curve) (Figure 6 and Table 2). The 18 months old title bio-formulation and its fresh counterpart demonstrate interesting behavior. The ongoing processes can be elucidated by comparing PVP-I_2_ (purple curve) with AV-PVP-Thyme-I_2_ (Figure 6 and Table 2).

UV-spectral analysis of PVP-I_2_ (purple curve) reveals iodide ions at 202 nm, molecular iodine at 205 nm, triiodide ions at 290 and 360 nm, and pentaiodide ions at 444 nm (Figure 6, purple curve) [29,30,34,35,36]. Once AV and Thyme are added to form AV-PVP-Thyme-I_2_ (red curve), the absorption pattern changes. The pentaiodide ions, which consist of actually (I_2_⋯I_3_^−^), disappear in the red curve (Figure 6 and Table 2). Adding the plant extracts leads to destabilization and loss of pentaiodide ions: [I_2_⋯I_3_^−^] → [I_2_] + [I_3_^−^](1)

Breaking apart into iodine molecules and triiodide ions. This alone should be a reason to record higher intensity absorptions of molecular iodine and triiodide anions in the red curve of the title bio-material. However, only higher intensity absorption of I_3_^−^ is observed in AV-PVP-Thyme-I_2_, while I_2_ and even I^−^ absorption bands show reduced intensities. This can be explained by: [I_2_] + [I^−^] ⇆ [I_3_^−^](2)

Iodine and iodide ions form more triiodide anions, hence again increasing the triiodide absorption intensities and at the same time decreasing their own. In this case, the concentration of molecular iodine should not decrease much because there was already a surplus in the sample. However, the redox reaction:Oxidation: [-C-O^−^] ⇆ [electron^−^] + [-C=O](3)
Reduction: [I_2_] + [electron^−^] ⇆ 2[I^−^](4)

May explain how the surplus of molecular iodine is used up to produce iodide ions, which will in turn increase the triiodide concentration, according to Equation (2). The oxidation reaction in Equation (3) refers to the hydroxyl groups in the AV- and Thyme bio-compounds. Accordingly, -C=O vibrational absorption intensities increase after adding AV- and Thyme extracts to the FTIR analysis, confirming the assumption in Equation (3).

Initially, AV- and Thyme bio-constituents could compete against triiodides being complexed by PVP through hydrogen bonding. Suitable plant biocomponents may interfere and hydrogen bond instead of triiodide ions, leading to their release from PVP. This would hamper the antimicrobial properties of the title formulation. The results of the antimicrobial testing are remarkable and even show higher inhibitory action after a storage period of 18 months. In conclusion, the competition between AV- and Thyme bio-compounds in AV-PVP-Thyme-I_2_ does not scale down the inhibitory action in the fresh sample or after storing it for 18 months. PVP protects triiodide ions within AV-PVP-Thyme-I_2_ through hydrogen bonding and acts as a sustained-release reservoir by inhibiting I_3_^−^ decomposition [34,35,36,128]. In contrast, this would have resulted in iodine release, decolorization of the formulation, and reduced antimicrobial properties after 18 months of storage. Instead, our results from the UV-spectral analysis and the antimicrobial testing confirm the absence of decolorization of the formulation; however, only a decrease in iodine and iodide ion concentrations and augmented antimicrobial properties were found.

Another interesting fact is related to the vibrational band absorptions of -C=O in PVP at 210, 218, and 221 (C=O) nm in the purple curve [34,35,36]. These shift to shorter wavelengths towards 208, 213, and 217 nm in combination with decreased absorption intensities in AV-PVP-Thyme-I_2_ (red curve), respectively (Figure 6a and Table 2). The blue shift with reduced absorption intensities implies more hydrogen bonding, encapsulation, and removal of conjugated systems by changing PVP-related C=O groups to C-O [34,35,36].

Thyme maceration extracts consist mainly of the phenolic monoterpenes thymol and carvacrol; the flavonoid luteolin; as well as the phenolic acids trans-rosmarinic acid, syringic acid, and gallic acid in descending order [90,91,92,93]. Keeping in mind the overlapping of the absorption peaks with the triiodide moieties, assigning all expected Thyme maceration extract components is not possible. Nevertheless, we compared our samples with the UV spectra of thymol and carvacrol in Table 2 Our formulations contain similar absorption peaks at 202, 204, 207, 210, 212, 277, and 415 nm, indicating the availability of thymol and carvacrol (Table 2). Trans-rosmarinic acid is confirmed by the peaks at 206, 250, and 330 nm, in accordance with previous investigations [90,91,92,93,94].

As a conclusion, AV- and Thyme polyphenols are not able to remove triiodide ions from PVP through hydrogen bonding completely. Triiodide ions are protected by PVP even after 18 months within the formulation without releasing them. The analytical results show no change in the composition, while the antimicrobial testing showed even higher inhibition by the 18 months older sample in comparison to the fresh AV-PVP-Thyme-I_2_. Therefore, the longer the sample is stored, the more triiodide ions are available and encapsulated by PVP within a sustained-release reservoir [34,35,36,128].

#### 2.2.3. Fourier-Transform Infrared (FTIR) Spectroscopy

FTIR analysis of Thymol, PVP-I_2_, and AV-PVP-Thyme-I_2_ confirms the purity of the title formulation and its composition (Figure 7 and Appendix A).

Figure 7 reveals similarities in the FTIR spectrum of AV-PVP-Thyme-I_2_ and PVP-I_2_, in comparison to our previous investigations [34,35,36]. The addition of AV and Thyme extract leads to a broadening of the bands between 3700 and 3500 cm^−1^ due to an increase in hydrogen bonding by the added AV- and Thyme-biocomponents through their hydroxyl- and carboxylic acid groups (Figure 7). The FTIR spectrum also shows higher absorption intensities between 3300 and 3100, 1850 and 1450 cm^−1^, and in the fingerprint region (Figure 7). The band related to asymmetric -C=O stretching vibrations at 1658 cm^−1^ in PVP-I_2_ (B, purple curve) is blue shifted towards 1653 cm^−1^ (A, red curve), with an additional broadening and increased absorption intensity in AV-PVP-Thyme-I_2_ (Figure 7 and Table 3) [34,35,36].

These findings indicate that adding AV- and Thyme bio-compounds increased hydrogen bonding with the carbonyl groups of the PVP in competition to the complexed polyiodide moieties. This process is enabled by the release of PVP-complexed pentaiodide ions and their subsequent decomposition into triiodide ions and iodine molecules, according to Equation (1). Additionally, increased absorption of the carbonyl stretching vibration confirms the availability of the newly added -C=O groups within the AV- and Thyme extracts originating from aloin, acemannan, and trans-rosmarinic acid originating from carboxylic -COO and -C(=O)OCH_3_ ester groups (Figure 7 and Table 3) [34,35,36,43,44,58,91,92]. At the same time, an AV-based –OH stretching vibration from aloin, emodin, galacturonic acid, and mannose is available in AV-PVP-Thyme-I_2_ at 3445 cm^−1^, in agreement with previous studies (Table 3) [34,35,36,44,45]. After adding AV and Thyme, the asymmetric and symmetric C-H stretching vibrations are red-shifted from 2947 to 2955 cm^−1^ and from 2862 to 2868 cm^−1^, respectively (Table 3) [34,35,36,44]. This finding confirms that incorporating AV and Thyme phenolic compounds like aloin, aloe emodin, rosmarinic acid, thymol, and carvacrol into the formulation increases the number of functional group conjugations and the solvent effect [34,35,36,43]. The same is witnessed in the region between 3260 and 3130 cm^−1^ with new, high-intensity absorption bands at 3445, 3383, 3215, and 3148 cm^−1^. These bands correspond to the hydroxyl and carboxylic groups of the bio-extracts Thyme and AV in AV-PVP-Thyme-I_2_ (Figure 7 and Table 3).

The low absorption band at 1753 cm^−1^ in both samples reveals the low degree of acetylation of the -C=O groups asymmetrical stretching vibration in PVP. These vibrations indiscriminately increase in intensity when AV- and Thyme extracts are added (Figure 7 and Table 3) [34,35,36]. The vibration of the -C=O group in the polymeric PVP matrix is assigned to the band at 1038 cm^−1^ in PVP-I_2_ (Figure 7 and Table 3). Once AV- and Thyme extracts are added, this band is seen with less intensity and blue shifted towards 1036 cm^−1^ (Figure 7 and Table 3). Reduced absorption intensity coupled with a small blue shift indicates higher complexation around the PVP-carbonyl group [34,35,36]. Another encapsulation process affects the bands of asymmetric stretching vibrations for -C-H groups and -CH_2_ twisting at 2990 and 880 cm^−1^, which show up with reduced intensities at 2990 and 881 cm^−1^ after adding AV- and Thyme-extracts, respectively [34,35,36].

In conclusion, incorporating AV- and Thyme extracts into PVP-I_2_ resulted in broadening and increased absorption intensities by their -C=C-, -C=O-, and -COOH- groups in the formulation. Hydrogen bonding increased with the PVP polymetric matrix, which releases pentaiodide ions as part of the process. The PVP backbone composed of [–CH-CH_2_-]_n_- groups is more coiled through encapsulation of triiodide ions and further AV- and Thyme bio-compounds [34,35,36]. The complexation of triiodide anions increases over time and results in a PVP polymeric matrix entangled or coiled around the triiodide moieties, acting as a protective, sustained-released reservoir around them [34,35,36,128]. The coiled or entangled structure is confirmed in the SEM images of the bandages and the face mask in the form of small, circular, or cube-like patches covering the surface of the tissues (Figure 3d and Figure 4b).

### 2.3. X-ray Diffraction (XRD)

The XRD analysis of AV-PVP-Thyme-I_2_ confirms the purity of the sample with peaks from AV, Thyme, PVP, and iodine (Figure 8).

The XRD analysis reveals almost the same peaks for AV-PVP-Thyme and AV-PVP-Thyme-I_2_. AV-PVP-Thyme-I_2_ reveals clearly lower intensities in the XRD pattern. The only peak with slightly higher intensity in the iodinated compound in relation to AV-PVP-Thyme is at 2θ = 30° (Figure 8). Previously, other investigators assigned 2Theta values around 24, 30, 37, and 46° to iodine, which confirms the composition of our title bio-material (Figure 8 and Table 4) [120,121].

The decrease in weak intensities after iodination indicates strong encapsulation of iodine into the PVP backbone in the form of triiodide anions. The only shifts in maxima occur at 2θ = 45.81° and 45.92° in AV-PVP-Thyme (green curve) towards 2θ = 45.84° and 45.95° after iodination (red curve), indicating higher complexation of polyiodide moieties into the formulation AV-PVP-Thyme-I_2_ (Figure 8 and Table 4) [81].

Additionally, the absence of new peaks in AV-PVP-Thyme-I_2_ points to an amorphization of iodine during this process, as witnessed before in our previous study of AV-PVP-Sage-I_2_ (Figure 8 and Table 4) [34].

Thymol appears in the XRD analysis at 2Theta values of around 14.92, 14.98, 24.43, 30.08, and 30.14°, in agreement with other investigations (Table 4) [34,35,81]. The peaks at 2θ = 14.89 and 23.37° are assigned according to previous studies to PVP (Table 4) [34,35]. Finally, AV-bio-compounds can be confirmed through the peaks around 2θ = 46, 50, and 63° (Table 4) [34,35,81].

The XRD analysis confirmed the purity of AV-PVP-Thyme-I_2_ (Figure 8 and Table 4).

### 2.4. Antimicrobial Activities of AV-PVP-Thyme-I_2_

The inhibitory action of the title bio-material AV-PVP-Thyme-I_2_ was tested by disc diffusion assay (DD) against 10 reference microorganisms on sterile discs at concentrations of 11, 5.5, and 2.75 µg/mL. We impregnated sterile, braided, surgical PGA sutures, cotton bandages, surgical face masks, and KN95 face masks with a concentration of 11 µg/mL. The reference microorganisms included the fungus *C. albicans* WDCM 00054 Vitroids, the Gram-positive bacteria *S. pneumonia* ATCC 49619, *S. aureus* ATCC 25923, *S. pyogenes* ATCC 19615, *E. faecalis* ATCC 29212, and *B. subtilis* WDCM0003, as well as the Gram-negative *E. coli* WDCM 00013 Vitroids, *P. mirabilis ATCC 29906*, *P. aeruginosa* WDCM 00026 Vitroids, and *K. pneumonia* WDCM00097 Vitroids. The negative controls were ethanol and water, which had no inhibitory effect. These are not included in Table 5.

The common antibiotics gentamycin (G) and nystatin (NY) were utilized as positive controls (Table 5). After storing the title formulation for 18 months, we performed DD tests again and obtained surprisingly augmented antimicrobial properties in comparison to the fresh AV-PVP-Thyme-I_2_. Storing the sample did not result in the decomposition of the triiodide ions. During this time, the triiodide ions were protected by the polymeric PVP matrix, which acted successfully as a sustained-release reservoir. Such complexation prevents the decomposition and subsequent loss of iodine molecules before confronting the cell membranes of microorganisms (Table 5) [128].

Table 5 is arranged from up to down in decreasing inhibitory action against the 10 reference microorganisms. Accordingly, the pathogens were susceptible in decreasing order towards *C. albicans* WDCM 00054, followed by the Gram-positive bacteria (*S. aureus* ATCC 25923, *B. subtilis* WDCM 00003, *S. pyogenes* ATCC 19615, *E. faecalis* ATCC 29212, *S. pneumoniae* ATCC 49619), and lastly the Gram-negative pathogens (*K. pneumoniae* WDCM 00097, *E. coli* WDCM 00013, *P. aeruginosa* WDCM 00026, and *P. mirabilis* ATCC 29906) (Table 5). The DD tests revealed the highest ZOI for *C. albicans* WDCM 00054 and confirmed that AV-PVP-Thyme-I_2_ is a strong antifungal agent. Our bio-formulation encompasses even the control antibiotic nystatin (NY) almost fivefold, with 61 mm compared to 16 mm, respectively (Table 1). The remarkable ZOI of 61 mm in the 18 months old sample is clearly an indicator of enhanced antimicrobial properties compared to the 50 mm ZOI in the fresh AV-PVP-Thyme-I_2_ (Table 5). The DD studies against *C. albicans* WDCM 00054 resulted in ZOI = 50, 35, 30, and 15 for the concentrations of 11, 5.5, 2.75, and 1.38 µg/mL, respectively (Figure 9 and Table 5).

The disc diffusion studies showed the same general order of susceptibility to the pathogens in Table 5. We had the same trends for the formulation AV-PVP-Sage-I_2_ [34]. *S. aureus ATCC 25923* shows intermediate inhibition zones of 18, 17, and 15 mm against the title compound and 20, 15, and 14 mm for AV-PVP-Sage-I_2_ (Figure 9b and Table 5) [34]. The Gram-negative *K. pneumoniae* WDCM 00097 is more susceptible with ZOI = 13, 12, and 10 mm than the other Gram-negative pathogens (Figure 9c and Table 5). The same happens in the formulation AV-PVP-Sage-I_2_ with 13 and 9 mm [34]. This trend is related to the motility of the pathogens. Accordingly, the most motile, swarming bacteria *P. mirabilis* ATCC 29906 is not inhibited by AV-PVP-Thyme-I_2_ in any of the tests except when impregnated into a KN95 mask (Table 5). The bio-formulation with Sage does not inhibit these motile bacteria as well. The only difference between the formulations AV-PVP-Thyme-I_2_ and AV-PVP-Sage-I_2_ is their inhibitory action towards *P. aeruginosa* WDCM 00026 (Table 5) [34]. The motile bacillus *P. aeruginosa* WDCM 00026 is resistant to AV-PVP-Sage-I_2_ [34]. The bio-hybrid AV-PVP-Thyme-I_2_ exerts intermediate inhibitory action against *P. aeruginosa* WDCM 00026 with ZOI = 11, 10, and 9 mm. Another difference is seen in the susceptibility of *B. subtilis* WDCM 00003 to both bio-formulations. The inhibition of *B. subtilis* WDCM 00003, a rod-shaped bacilli, is higher in the title formulation AV-PVP-Thyme-I_2_ compared to AV-PVP-Sage-I_2_ with ZOI = 19, 15, and 14 mm against Z = 13, 12, and 11 mm, respectively (Table 5) [34]. In general, the Thyme formulation has higher inhibitory action than the Sage [76,77,78,79]. This could be a result of the availability of the monoterpenes Thymol and Carvacrol within the title compound AV-PVP-Thyme-I_2_ [76,77,78,79,80,81,82,83,84,85,86,87,88,89,90,91,92]. These monoterpenes are known for their antimicrobial actions against many microorganisms [80,81,82,83,84,85,86,87,88,89,90,91,92].

As a conclusion, the bio-hybrid AV-PVP-Thyme-I_2_ is a strong antifungal agent against *C. albicans* WDCM 00054. The Gram-positive bacteria *S. aureus* ATCC 25923 is also inhibited strongly in comparison to the common antibiotic gentamycin, while *B. subtilis* WDCM 00003 and *S. pyogenes* ATCC 19615 are intermediately susceptible. These findings enable the potential use of our title bio-formulation as an antifungal and antibacterial coating, as well as a surface contact agent against the mentioned pathogens.

Surgical sutures are very important tools in surgery to close open wounds [32,34,36,69]. Especially biodegradable, braided PGA sutures are used in the medical field, including oral surgery [32,34,36,69]. Resistant pathogens and/or conditions favoring the proliferation of microorganisms can lead to surgical-site infections, which delay or impede the healing process [32,34,36,69]. Antimicrobial coatings may reduce the incidence of surgical-site infections, ameliorate wound closure, and ease patient suffering [32,34,36,69]. Therefore, we studied the inhibitory effects of dip-coated surgical PGA sutures (S) (Table 5). The highest susceptibility to the title biohybrid in surgical sutures is recorded for the Gram-positive bacteria *E. faecalis* ATCC 29212 (4 mm) (Table 5). *C. albicans* WDCM 00054, *S. aureus ATCC 25923*, and *B. subtilis* WDCM 00003 also show inhibition zones of 3 mm on surgical sutures (Figure 10 and Table 5).

Further inhibited Gram-positive bacteria include *S. pyogenes* ATCC 19615 (2 mm) and *S. pneumoniae* ATCC 49619 (1 mm) (Table 5). *K. pneumoniae* WDCM 00097 is the only susceptible Gram-negative pathogen on dip-coated surgical sutures with a ZOI of 2 mm (Table 5 and Figure 10c).

As a conclusion, surgical PGA sutures coated with our bio-formulation AV-PVP-Thyme-I_2_ may have potential as antimicrobial agents to prevent surgical-site infections against *C. albicans* WDCM 00054 and Gram-positive bacteria *E. faecalis* ATCC 29212, *S. aureus ATCC 25923*, and *B. subtilis* WDCM 00003.

Face masks were confirmed as essential tools in mitigating opportunistic pathogens during the COVID-19 pandemic [2,3,4,36]. They are pivotal as personal protective equipment to prevent the spread of microbes during normal flu or cold seasons in all public settings, including clinics, hospitals, elderly homes, emergency wards, and intensive care units [2,3,4,36]. The last COVID-19 outbreak was marked by an inability to provide the public with high-quality face masks globally [2,3,4,36]. Supply chains broke down, pharmacies and drug stores ran out of stocks, prices skyrocketed, while quality, safety, sustainability, and environmental concerns remained [2,4,36]. Low-income communities worldwide had no adequate access to face masks. Re-using face masks by spraying them with antimicrobial surface contact agents can mitigate microbial proliferation, reduce environmental pollution, increase accessibility, and increase sustainability.

We coated surgical face masks (M) and KN95 masks with our bio-hybrid AV-PVP-Thyme-I_2_ and studied the inhibitory action (Table 5.) The results reveal the same trends mentioned for discs, with the best results for *C. albicans* WDCM 00054, followed by strong inhibitory action against Gram-positive and intermediate susceptibility of Gram-negative bacteria (Table 5). *C. albicans* WDCM 00054 and the Gram-positive bacteria are highly susceptible against AV-PVP-Thyme-I_2_ coated on KN95 face masks, then the white surgical face mask layer (M^W^), and lastly the blue face mask layer (M^B^) (Table 5). The inhibitory action against *C. albicans* WDCM 00054 is ameliorated when AV-PVP-Thyme-I_2_ is impregnated on the KN95 mask (80 mm, Figure 11d), followed by the surgical facemask white layer (55 mm), and lastly the blue layer (45 mm) (Table 5 and Figure 11).

Among the Gram-positive reference strains, the antimicrobial action of AV-PVP-Thyme-I_2_ is strongest on KN95 against *B. subtilis* WDCM00003 with 40 mm, then *S. aureus* ATCC 25932 with 38 mm, *S. pyogenes* ATCC 19615 with 30 mm, *E. faecalis* ATCC 29212 with 28 mm, and lastly *S. pneumoniae* ATCC 49619 with 26 mm (Table 5 and Figure 11). The title formulation exerted strong inhibitory properties in comparison to the common antibiotics against Gram-negative bacteria, but with mixed results (Table 5). *K. pneumoniae* WDCM 00097 was more susceptible to the title bio-formulation on M^B^ with 28 mm, followed by KN95 with 24 mm and M^W^ with 21 mm (Table 5 and Figure 11). *E. coli* WDCM 00013 and *P. aeruginosa* WDCM 00026 showed both higher susceptibilities on M^W^ with ZOI = 25 in comparison to M^B^ and KN95 with 24/16 and 20/23 mm, respectively (Table 5).

As a conclusion, the bio-hybrid AV-PVP-Thyme-I_2_ exerts strong inhibitory action on KN95 face masks and surgical face masks against *C. albicans* WDCM 00054, the Gram-positive microorganisms *B. subtilis* WDCM 00003, *S. aureus* ATCC 25923, *S. pyogenes* ATCC 19615, *E. faecalis* ATCC 29212, and *S. pneumoniae* ATCC 49619. Gram-negative pathogens *K. pneumoniae* WDCM 00097, *E. coli* WDCM 00013, and *P. aeruginosa* WDCM 00026 were more susceptible to the surgical face masks. *P. mirabilis* ATCC 29906 is only inhibited when AV-PVP-Thyme-I_2_ is impregnated on KN95 masks. Antiviral tests were not performed and are planned to be investigated in future studies in our group.

Wound treatment needs a sophisticated approach to hasten the healing process successfully by preventing infection of the wound surroundings [15,36,39,40,42,46,48,52]. Impregnating cotton bandages with antimicrobial substances may mitigate the proliferation of harmful microorganisms on the wound and reduce treatment duration [15,39,40]. Therefore, we coated sterile cotton bandages (B) with our bio-formulation AV-PVP-Thyme-I_2_ and studied its action against our reference strains (Table 5). As expected, the same pattern of AV-PVP-Thyme-I_2_ antimicrobial action is exerted, with the highest antimicrobial action towards *C. albicans* WDCM 00054, followed by Gram-positive and Gram-negative pathogens (Table 5). *C. albicans* WDCM 00054 reveals the highest inhibition zone (53 mm) in comparison to all other reference strains and the common antibiotics (Table 5 and Figure 12).

The Gram-positive pathogens are also highly susceptible to impregnated bandages with a ZOI = 35, 32, 28, 24, and 23 mm against *B. subtilis* WDCM 00003, *S. aureus* ATCC 25932, *S. pyogenes* ATCC 19615, *S. pneumoniae* ATCC 49619, and *E. faecalis* ATCC 29212, respectively (Figure 12 and Table 5).

Even the Gram-negative bacteria showed high-to-intermediate inhibitory action. *K. pneumoniae* WDCM 00097 was highly susceptible with 28 mm ZOI, followed by *E. coli* WDCM 00013 with 23 mm and *P. aeruginosa* WDCM 00026 with 20 mm, respectively (Figure 12 and Table 5).

As a conclusion, cotton bandages enhance the antimicrobial properties of AV-PVP-Thyme-I_2_ against all the selected reference strains. The highest inhibition zones were recorded against *C. albicans* WDCM 00054, suggesting its use as an antifungal agent on bandage as well as on discs, face masks, and sutures. Our title bio-formulation has potential applications against Gram-negative and Gram-positive bacteria on face masks, bandages, discs, and sutures in descending order.

The general susceptibility pattern remains valid for all the tested carrier materials (suture, disc, bandage, surgical face mask, and KN95). The material, which allows a dense, homogenous coating by small, spherical, or cube-like patches of AV-PVP-Thyme-I_2_, seems to be the most suitable carrier, enhancing the antimicrobial properties. Table 5 is arranged in order of highest to lowest susceptibility, starting with *C. albicans* WDCM 00054, Gram-positive, and finally Gram-negative microorganisms. The arrangement of Gram-positive pathogens is ruled by different factors in accordance with our previous studies [34,35,36]. Morphology, size, properties, structure, and motility of the microorganisms are a function of their susceptibility to our bio-hybrid AV-PVP-Thyme-I_2_. The main reason is the iodine release from the sustained-released reservoir PVP, followed by the antimicrobial actions of the monoterpenes Thymol and Carvacrol (Figure 13).

These monoterpenes and AV components are complexed by PVP through hydrogen bonding. The analytical results suggest a firm PVP-I_2_ complex even after a storage period of 18 months and point at the availability of smart triiodides protected within the PVP-sustained reservoir over that long period.

Molecular iodine is a small, lipophilic molecule. This property enables its diffusion through the cell membranes of the studied microorganisms. Phenolic acids originating from the AV- and Thyme extract pass also through the cell membranes. Gram-positive pathogens, with their lipophilic peptidoglycan layers, accelerate diffusion and cell death. Gram-negative pathogens are affected less by molecular iodine and phenolic acids. Nevertheless, depending on the carrier (KN95, surgical face mask, and bandage), high inhibition zones are recorded. However, porin channels in Gram-negative bacteria allow hydrophilic triiodide ions and iodide ions, as well as small phenolic acid diffusion through the outer membranes. However, the porin channel diffusion is successful in KN95, surgical face masks, and bandages. Cell membrane diffusion remains the major mechanism ameliorating Gram-positive bacteria inhibition on KN95, surgical face masks, and bandages as well.

The morphology of the microorganisms characterizes their inhibition by the title bio-formulation AV-PVP-Thyme-I_2_. The susceptibility of the Gram-positive pathogens is highest in *S. aureus* ATCC 25932, which consists of clusters of non-motile cocci. The pattern continues with non-motile chains and pairs of cocci. Gram-negative bacteria inhibition is directly proportionate to their motility. The swarming bacteria *P. mirabilis* ATCC 29906 is only inhibited on KN95. Therefore, homogenously coated carriers with small, spherical, cube-like patches of AV-PVP-Thyme-I_2_ ensure even inhibition of motile Gram-negative bacteria.

In general, the results on discs, surgical PGA sutures, cotton gauze bandages, and face masks enable the potential use of the title bio-formulation AV-PVP-Thyme-I_2_ in many applications. AV-PVP-Thyme-I_2_ may be utilized to prevent inflammatory processes, to curb microbial proliferation, to reduce wound and surgical-site infections, to prevent airborne transmission of microbes, and as a surface killing agent (Figure 14).

The biohybrid can be potentially used in oral care products or dental treatment to facilitate the removal of colonizing oral pathogens like *S. aureus*, *E. faecalis*, *S. pyogenes*, *E. coli*, *P. aeruginosa*, and *C. albicans*. These opportunistic pathogens form biofilms in oral cavities surrounding teeth and gingiva and can even reach the dentin walls of root canals [131,132,133] (Figure 15).

*C. albicans* and *E. faecalis* colonize oral cavities and form dental biofilm in and around teeth (Figure 14). The biofilm leads progressively to inflammation, plaque formation, dental caries, periodontal disease, and other dangerous systemic infections like rheumatoid arthritis, chronic kidney disease, and inflammatory bowel disease [131,132,133]. Throat and oral cavity infections caused by *S. pyogenes* from dental plaque can lead to serious illnesses like pneumonia, endocarditis, or encephalitis [133]. Such colonizers could be stopped by using our bio-formulation as mouthwash or gargle due to its high inhibitory action on *C. albicans* and further oral pathogens (Table 5 and Figure 14). Future investigations should include cytotoxicity, cell culture, in vivo, and antiviral studies to verify the suitability of the mentioned applications.

## 3. Materials and Methods

### 3.1. Materials

We harvested the AV leaves in December from the botanical garden of the Ajman University campus in Ajman, UAE. Liofilchem Diagnostici (Roseto degli Abruzzi (TE), Italy) delivered the disposable sterilized Petri dishes with Mueller Hinton II agar, the McFarland standard sets, gentamicin (9125, 30 µg/disc), and nystatin (9078, 100 IU/disc). We bought from Sigma-Aldrich Chemical Co. (St. Louis, MO, USA) the reference strains *E. coli* WDCM 00013 Vitroids, *K. pneumoniae* WDCM 00097 Vitroids, *P. aeruginosa* WDCM 00026 Vitroids, *Bacillus subtilis* WDCM 0003 Vitroids, and *C. albicans* WDCM 00054 Vitroids. We obtained from Liofilchem (Roseto degli Abruzzi (TE), Italy) *P. mirabilis* ATCC 29906, *S. aureus* ATCC 25923, *S. pyogenes* ATCC 19615, *E. faecalis* ATCC 29212, and *S. pneumoniae* ATCC 49619. Sigma Aldrich (St. Louis, MO, USA) also provided us with Mueller Hinton Broth (MHB), Sabouraud Dextrose broth, pure ethanol, iodine (≥99.0%), and polyvinylpyrrolidone (PVP-K-30). Himedia (Jaitala Nagpur, Maharashtra, India) delivered sterile filter paper discs with a 6 mm diameter. Sterile polyglycolic acid (PGA) surgical sutures from DAMACRYL (USP: 3-0, Metric: 2, 19 mm, 75 cm, DC3K19) were provided by General Medical Disposable (GMD), GMD Group A.S., Istanbul, Turkey. We obtained sterile cotton gauze bandages and surgical, disposable, 3-ply non-woven face masks, as well as KN95 face masks, from our local pharmacy (FOMED, Qianjiang City, Hubei Province, China). All the utilized reagents were of analytical grade and used as delivered under sterile conditions. Absolute ethanol and ultrapure water were utilized in all experiments.

### 3.2. Preparation of Aloe Vera (AV) Extract and Thyme Extract

Fresh Thyme plants were purchased in August from a local store in Jordan. They were washed several times with distilled water to remove dust and soil, and then twice with absolute ethanol. The leaves were dried for 1 h at ambient temperature, and then 100 mg of a fresh Thyme plant consisting of leaves and stems was cut into pieces. These pieces were immediately transferred into a sterile, brown glass bottle with a screw cap. We added 100 mL of absolute ethanol, sealed the glass bottle, and stored it at ambient temperature in darkness for 3 months as part of a maceration. The macerate extract was then stored in a fridge at 3 °C under darkness for further use.

The leaves of the *Aloe Vera* (*Aloe barbadensis* Miller) plant were harvested during December from the botanical garden of Ajman University in the morning hours [34,35,36]. Within 10 min, the leaves with lengths of 35 to 50 cm were washed with water, then rinsed many times with distilled water, followed by absolute ethanol, then several times with ultrapure water, and finally left to dry for 1 h. After being dried, the AV leaves were cut with a sterile knife to be able to collect the mucilaginous gel in a sterile beaker. Then, the gel was transferred to a sterile mixer, mixed for 20 min until the gel was homogenized, and finally centrifuged at 4000 rpm for 40 min (3K 30; Sigma Laborzentrifugen GmbH, Osterode am Harz, Germany). The supernatant had a light-yellow color, was transferred immediately into a sterile brown bottle with a screw cap, and was stored in the fridge at 3 °C in darkness.

### 3.3. Preparation of AV-PVP-Thyme-I_2_

AV-PVP-Thyme-I_2_ is prepared by a simple one-pot synthesis. For this, 2 mL of pure AV gel is filled into a sterile beaker. Then, we added 2 mL of a freshly prepared solution of 1 g polyvinylpyrrolidone K-30 (PVP) in 10 mL of distilled water at ambient temperature under continuous stirring. In this mixture, we added 2 mL of Thyme extract while stirring at ambient temperature. Finally, 2 mL of freshly prepared iodine solution (0.05 g of iodine in 3 mL of absolute ethanol) was added at ambient temperature with continuous stirring. The formulation AV-PVP-Thyme-I_2_ was immediately placed into a sterile glass sample tube with a screw cap and stored in the fridge for further use at 3 °C in darkness. The formulation was kept for 18 months under these conditions and characterized in comparison to the fresh sample.

### 3.4. Characterization of AV-PVP-Thyme-I_2_

AV-PVP-Thyme-I_2_ was analyzed by SEM/EDS, Raman spectroscopy, UV-vis, FTIR, and X-ray diffraction (XRD) analyses. The analytical results verified the purity and the morphology of the formulation.

#### 3.4.1. Scanning Electron Microscopy (SEM) and Energy-Dispersive X-Ray Spectroscopy (EDX)

Scanning electron microscopy (SEM) and energy-dispersive X-ray spectroscopy (EDS) analyses were performed using VEGA3 from TESCAN (Brno, Czech Republic) at 15 kV. One drop of the formulation AV-PVP-Thyme-I_2_ was diluted with distilled water, placed on the carbon-coated copper grid, and dried. This sample was gold-coated using the Quorum Technology Mini Sputter Coater. The SEM analysis allowed a glimpse of the morphology, while the EDS analysis confirmed the purity of the title formulation.

#### 3.4.2. UV-Vis Spectrophotometry (UV-Vis)

The biohybrid AV-PVP-Thyme-I_2_ was analyzed using a UV-Vis spectrophotometer model 2600i from Shimadzu (Kyoto, Japan) in the wavelength range of 195 to 800 nm.

#### 3.4.3. Raman Spectroscopy

The formulation AV-PVP-Thyme-I_2_ was analyzed at ambient temperature on a RENISHAW (Gloucestershire, UK), equipped with an optical microscope. The sample was transferred into a cuvette (1 cm × 1 cm) and placed in front of the laser beam with an excitation of 785 nm. The beam was directed within a spot diameter of 2micron onto the sample by a 50× objective of a confocal microscope. A CCD-based monochromator collected the scattered light with a spectral range between 50 and 3400 cm^−1^ with a spectral resolution of −1 cm^−1^, output power of 0.5%, and integration time of 30 s.

#### 3.4.4. Fourier-Transform Infrared Spectroscopy (FTIR)

The formulation AV-PVP-Thyme-I_2_ underwent a FTIR analysis in the range between 400 and 4000 cm^−1^ on an ATR IR spectrometer with a diamond window (Shimadzu, Kyoto, Japan). The liquid sample was diluted with absolute ethanol and analyzed by FTIR.

#### 3.4.5. X-ray Diffraction (XRD)

AV-PVP-Thyme-I_2_ was analyzed using XRD from BRUKER (D8 Advance, Karlsruhe, Germany) by Cu radiation with a wavelength of 1.54060, coupled two theta/theta, time/step of 0.5 s, and a step size of 0.03.

### 3.5. Bacterial Strains and Culturing

The antimicrobial testing of AV-PVP-Thyme-I_2_ was performed against a selection of 10 reference microbial strains (*C. albicans* WDCM 00054 Vitroids, *S. aureus* ATCC 25923, *S. pneumoniae* ATCC 49619, *E. faecalis* ATCC 29212, *S. pyogenes* ATCC 19615, *Bacillus subtilis* WDCM 0003 Vitroids, *K. pneumoniae* WDCM 00097 Vitroids, *E. coli* WDCM 00013 Vitroids, *P. aeruginosa* WDCM 00026 Vitroids, and *P. mirabilis* ATCC 29906). These reference strains were stored first at −20 °C, then inoculated by adding fresh microbes to Mueller Hinton Broth (MHB) and kept in the fridge at 4 °C until needed.

### 3.6. Determination of Antimicrobial Activities of AV-PVP-Thyme-I_2_

AV-PVP-TCA-I_2_ was tested against a selection of 10 microbial strains (*C. albicans* WDCM 00054 Vitroids, *S. aureus* ATCC 25923, *S. pneumoniae* ATCC 49619, *E. faecalis* ATCC 29212, *S. pyogenes* ATCC 19615, *Bacillus subtilis* WDCM 0003 Vitroids, *K. pneumoniae* WDCM 00097 Vitroids, *E. coli* WDCM 00013 Vitroids, *P. aeruginosa* WDCM 00026 Vitroids, and *P. mirabilis* ATCC 29906). The positive controls were the common antibiotics gentamicin and nystatin (for *C. albicans* WDCM 00054 Vitroids only). Pure ethanol and ultrapure water were the negative controls and did not inhibit anything. All the tests were repeated three times, and the average was presented in this study. The formulation was impregnated and also tested on sterile discs, sterile PGA sutures, cotton gauze bandages, surgical facemasks, and KN95 masks against the same reference strains.

#### 3.6.1. Procedure for Zone of Inhibition Plate Studies

We utilized the zone of inhibition plate method to study the susceptibility of the 10 reference microbial strains against AV-PVP-Thyme-I_2_ [134]. The selected bacterial reference strains were suspended in 10 mL of MHB and incubated for 2 to 4 h at a temperature of 37 °C. The fungus *C. albicans* WDCM 00054 was incubated at 30 °C in Sabouraud Dextrose broth. All microbial cultures were adjusted to the 0.5 McFarland standard. We used sterile cotton swabs to seed disposable, sterilized Petri dishes with MHA uniformly with 100 μL of microbial culture. After drying for 10 min the plates were ready for antimicrobial testing of AV-PVP-Thyme-I_2_.

#### 3.6.2. Disc Diffusion Method (DD)

We followed the antimicrobial testing as per the recommendations of the Clinical and Laboratory Standards Institute (CLSI) [135]. Sterile filter paper discs were impregnated for 24 h by 2 mL of AV-PVP-Thyme-I_2_ at concentrations of 11 µg/mL, 5.5 µg/mL, 2.75, and 1.38 µg/mL. Then, these discs were dried for 24 h under ambient conditions. The antibiotic discs of gentamycin and nystatin were used as positive controls. A ruler was utilized to measure the diameter of the zone of inhibition (ZOI) to the nearest millimeter, which is the clear area around the disc. No clear inhibition zone means that the reference microbial strain was not inhibited.

### 3.7. Preparation and Analysis of Impregnated Sutures, Cotton Gauze Bandages, Surgical Face Masks, and KN95 Masks with AV-PVP-TCA-I_2_ (11 µg/mL)

The sterile, braided surgical PGA sutures with a length of 2.5 mL were impregnated for 24 h in a 50 mL solution of AV-PVP-Thyme-I_2_ at ambient temperature. The blue sutures became brown-blue and were then dried for 24 h under ambient conditions. The cotton gauze bandages, surgical face masks, and KN95 masks were cut with a sterile scissor into square pieces measuring 5 cm x 5 cm. These squares were impregnated in 50 mL of AV-PVP-Thyme-I_2_ for 24 h and dried for 24 h at ambient temperatures. All the impregnated materials were tested by disc diffusion methods against the same selection of 10 reference microbial strains.

### 3.8. Statistical Analysis

We used SPSS software (version 17.0, SPSS Inc., Chicago, IL, USA) for the statistical analysis and represented the data in mean values. The statistical significance between groups was calculated by a one-way ANOVA. Values of *p* < 0.05 were treated as statistically significant.

## 4. Conclusions

AMR is a continuous threat to health and quality of life. Pathogens evolve by using different mechanisms to outsmart attempts to curb their proliferation. Multi-drug-resistant ESKAPE microorganisms (*Enterococcus faecium*, *Staphylococcus aureus*, *Klebsiella pneumoniae*, *Acinetobacter baumannii*, *Pseudomonas aeruginosa*, *Enterobacter* spp., and *Escherichia coli*) cause globally increasing rates of morbidity and mortality. New approaches are pivotal because they are cost-effective, sustainable, and facile. Natural antimicrobial formulations consisting of synergistic medicinal plant extracts seem to offer alternatives to AMR. Plants developed defense systems based on their synergistically combined plethora of bio-compounds. As an alternative approach, we add molecular iodine to the mixture of biocomponents. Iodine is a known microbicide, and until now, there have been no known resistance mechanisms to iodine. Therefore, adding iodine enhances the antimicrobial activities of the plant bio-materials originating from AV and Thyme. Iodine must be complexed within PVP to prevent its untimely and fast release, mitigate its stability, and ensure its long-term use. Polymeric complexes of PVP and iodine are already used in many products with cosmeceutical, oral health, and pharmaceutical backgrounds, as well as being excellent antimicrobial agents. Our group used AV gel and Thyme extracts together within the sustained-release reservoir PVP-I_2_. Our formulation, AV-PVP-Thyme-I_2_, is confirmed to be a strong antifungal agent against *C. albicans* WDCM 00054 on sterile discs, sutures, cotton gauze bandages, surgical face masks, and KN95 face masks. Five Gram-positive microorganisms, including the notorious *S. aureus* ATCC 25923, were strongly susceptible to the title bio-formulation even at a low concentration of 11 µg/mL, especially on cotton gauze bandages and face masks. Three out of four Gram-negative pathogens, including reference strains of the dangerous ESKAPE pathogens *K. pneumoniae* WDCM 00097, *P. aeruginosa* WDCM 00026, and *E. coli* WDCM 00013, were strongly inhibited by the same low concentration of 11 µg/mL AV-PVP-Thyme-I_2_. *P. mirabilis* ATCC 29906 was only susceptible with a ZOI of 25 mm on KN95 masks. Face masks and cotton bandages allow homogenous coating by AV-PVP-Thyme-I_2_ with small, spherical, cube-like patches and guarantee the high susceptibility of the selected reference strains.

The microbicidal formulation AV-PVP-Thyme-I_2_ could be potentially applied in oral treatments and surgery because it inhibits the oral cavity-related opportunistic pathogens (*S. aureus* ATCC 25923, *E. faecalis* ATCC 29212, *S. pyogenes* ATCC 19615, *E. coli* WDCM 00013, *P. aeruginosa* WDCM 00026, and *C. albicans* WDCM 00054) colonizing oral cavities, which in the worst-case scenario lead to dangerous systemic infections. They also cause throat and oral cavity infections, resulting in potentially fatal illnesses like pneumonia and others. Further applications could be skin antiseptics or inanimate surface disinfectants against community- and hospital-acquired infections. Impregnated face masks with AV-PVP-Thyme-I_2_ strongly inhibited all 10 reference strains. This application could enable the re-use of face masks as a sustainable solution for the planet and low-income populations. The results on cotton gauze bandages encourage the use of AV-PVP-Thyme-I_2_ in wound dressing materials. Further investigations are needed to evaluate the in vivo biological activities.

## Figures and Tables

**Figure 1 ijms-25-01133-f001:**
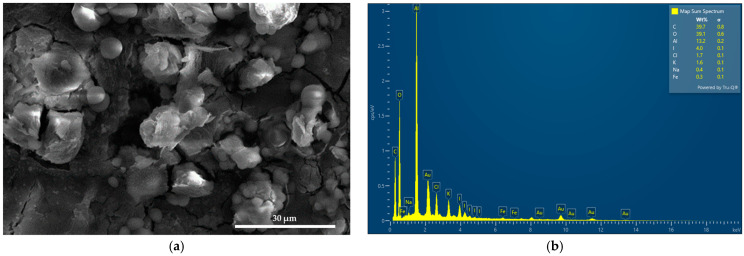
Scanning electron microscopy (SEM) (**a**) and energy-dispersive spectroscopy (EDS) (**b**) of AV-PVP-Thyme-I_2_.

**Figure 2 ijms-25-01133-f002:**
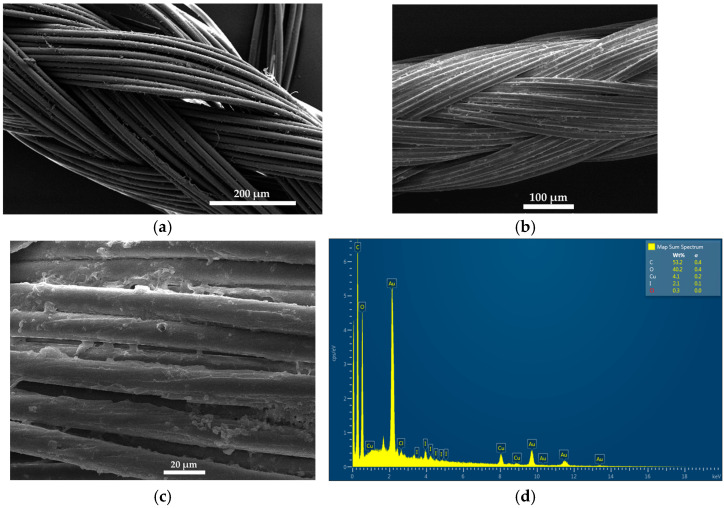
Scanning electron microscopy (SEM) of surgical sutures impregnated with AV-PVP-Thyme-I_2_: PGA suture at: (**a**) plain PGA suture [34], (**b**) 100 µm, and (**c**) 20 µm. (**d**) Energy-dispersive spectroscopy (EDS) of coated sutures.

**Figure 3 ijms-25-01133-f003:**
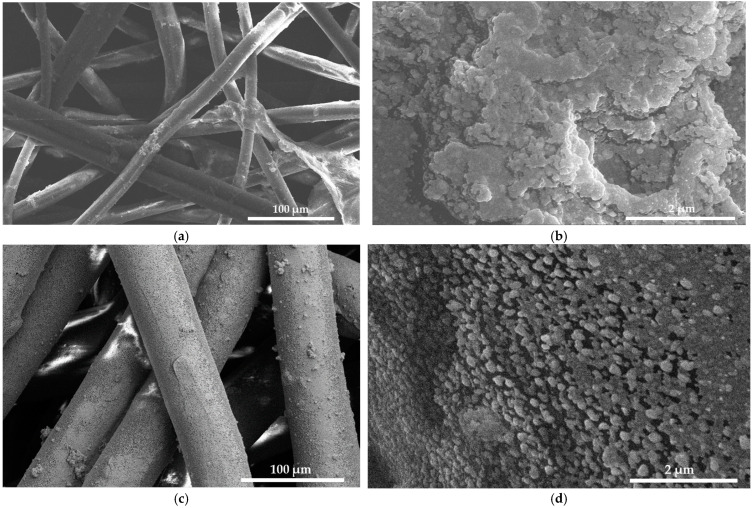
Scanning electron microscopy (SEM) of surgical face masks dip-coated with AV-PVP-Thyme-I_2_: dense, white inner layer (**a**); general appearance at 100 µm; (**b**) detailed view at 2 µm; net, blue outer layer; (**c**) general structure at 30 µm; (**d**) detailed view at 2 µm.

**Figure 4 ijms-25-01133-f004:**
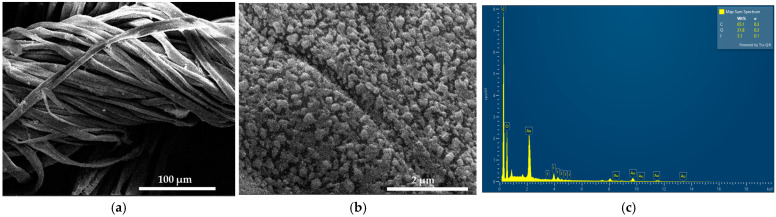
Scanning electron microscopy (SEM) of cotton surgical bandages impregnated with AV-PVP-Thyme-I_2_: bandage at: (**a**) 100 µm and (**b**) 2 µm. (**c**) Energy-dispersive spectroscopy (EDS) of coated bandages.

**Figure 5 ijms-25-01133-f005:**
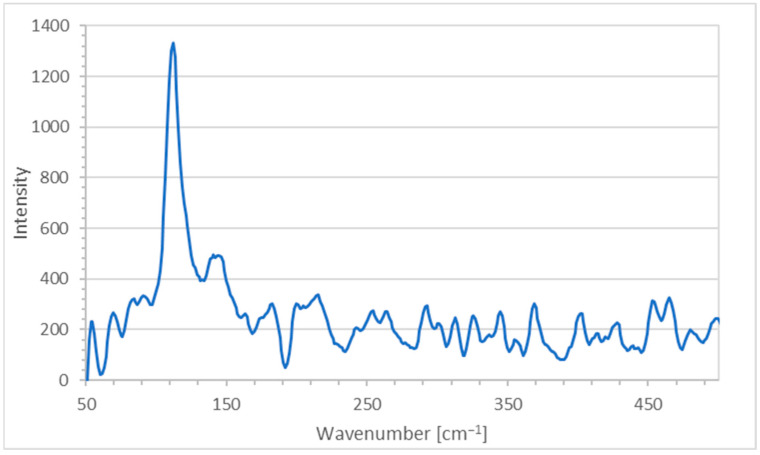
Raman spectroscopic analysis of AV-PVP-Thyme-I_2_.

**Figure 6 ijms-25-01133-f006:**
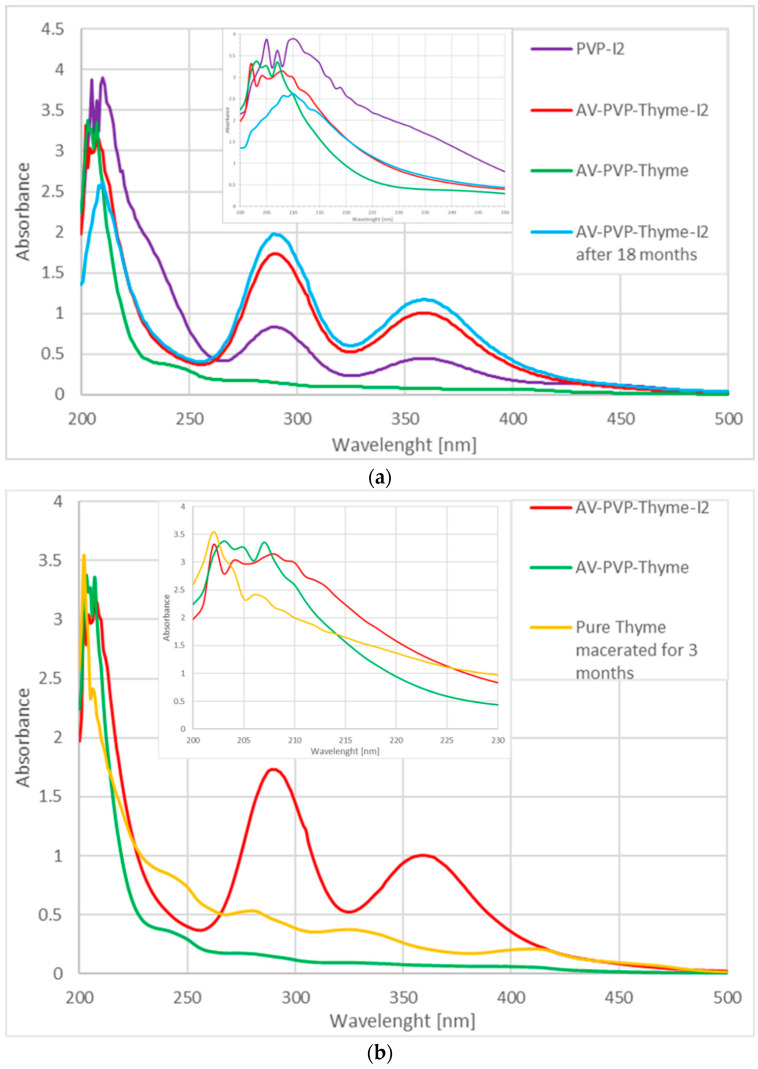
UV-vis analysis of AV-PVP-Thyme, AV-PVP-Thyme-I_2_, AV-PVP-Thyme-I_2_ (after 18 months), pure Thyme macerated for 3 months, and PVP-I_2_ (200–500 nm): UV-vis spectrum of AV-PVP-Thyme-I_2_ (**a**) in comparison to PVP-I_2_ with inlay, (**b**) in comparison to pure Thyme macerated for 3 months with inlay, and (**c**) between 200 and 450 nm only (AV-PVP-Thyme: green; AV-PVP-Thyme-I_2_: red; AV-PVP-Thyme-I_2_ after 18 months of storage: blue; PVP-I_2_: purple; and pure Thyme macerated for 3 months: orange).

**Figure 7 ijms-25-01133-f007:**
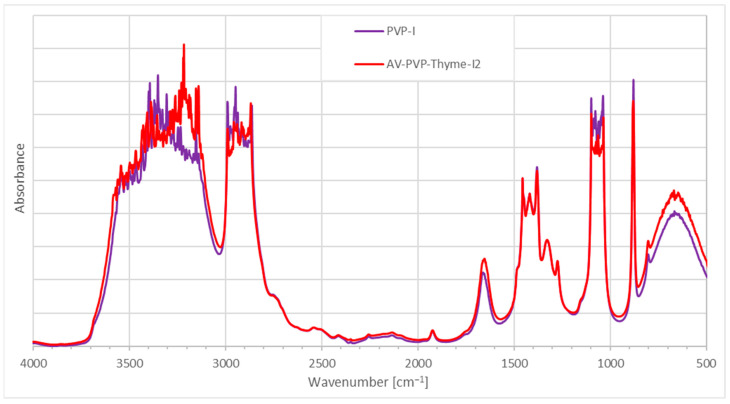
Fourier-transform infrared (FTIR) spectroscopic analysis of AV-PVP-Thyme-I_2_ (red) and PVP-I (purple).

**Figure 8 ijms-25-01133-f008:**
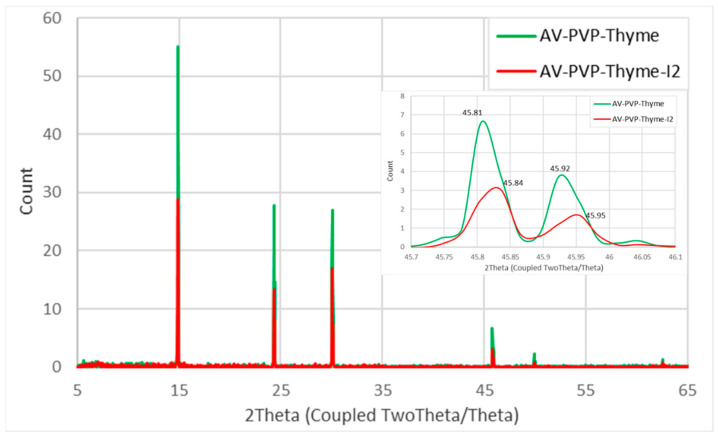
X-ray diffraction (XRD) analysis of AV-PVP-Thyme-I_2_.

**Figure 9 ijms-25-01133-f009:**
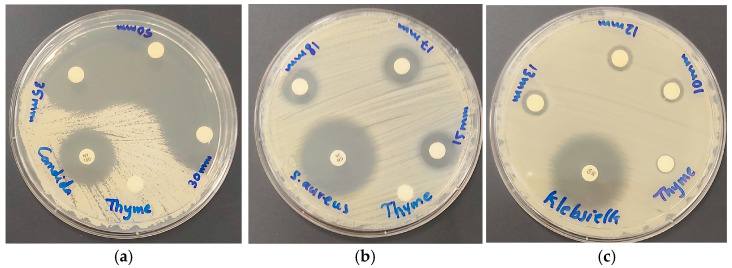
Disc diffusion assay of AV-PVP-Thyme-I_2_ (with concentrations of 11, 5.5, and 2.75 µg/mL) with the positive control antibiotics nystatin (100 IU) and gentamicin (30 µg/disc). From left to right: AV-PVP-Thyme-I_2_ against (**a**) *C. albicans* WDCM 00054, (**b**) *S. aureus* ATCC 25932, and (**c**) *K. pneumoniae* WDCM 00097.

**Figure 10 ijms-25-01133-f010:**
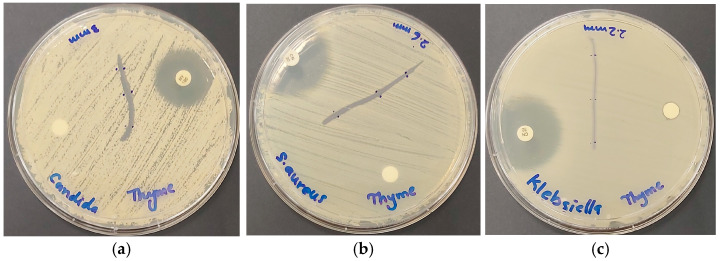
Impregnated, sterile PGA sutures with AV-PVP-Thyme-I_2_, with the positive control antibiotics nystatin (100 IU) and gentamicin (30 µg/disc). From left to right: AV-PVP-Thyme-I_2_ (11 µg/mL) against (**a**) *C. albicans* WDCM 00054, (**b**) *S. aureus* ATCC 25932, and (**c**) *K. pneumoniae* WDCM 00097.

**Figure 11 ijms-25-01133-f011:**
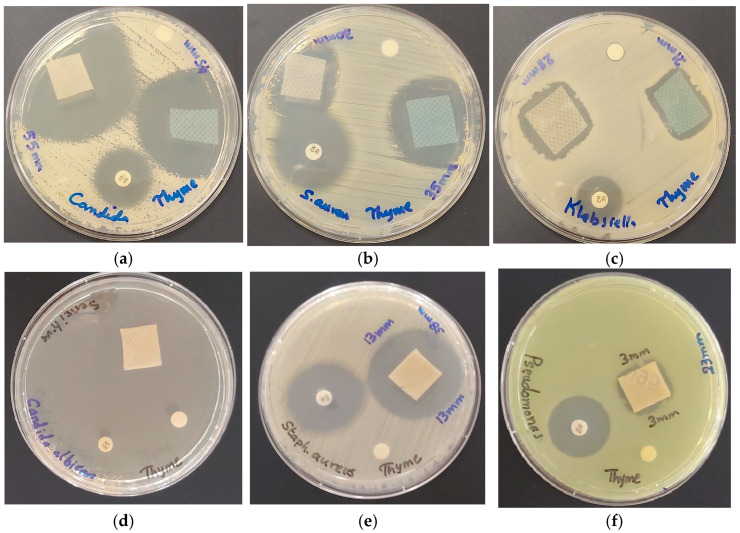
Impregnated sterile mask tissues with AV-PVP-Thyme-I_2_, with the positive control antibiotics nystatin (100 IU) and gentamicin (30 µg/disc). From left to right: AV-PVP-Thyme-I_2_ (11 µg/mL) against surgical face mask tissues blue and white (**a**) *C. albicans* WDCM 00054, (**b**) *S. aureus* ATCC 25932, (**c**) *K. pneumoniae* WDCM 00097 and AV-PVP-Thyme-I_2_ (11 µg/mL) against KN95 and (**d**) *C. albicans* WDCM 00054, (**e**) *S. aureus* ATCC 25932, (**f**) *P. aeruginosa* WDCM 00026.

**Figure 12 ijms-25-01133-f012:**
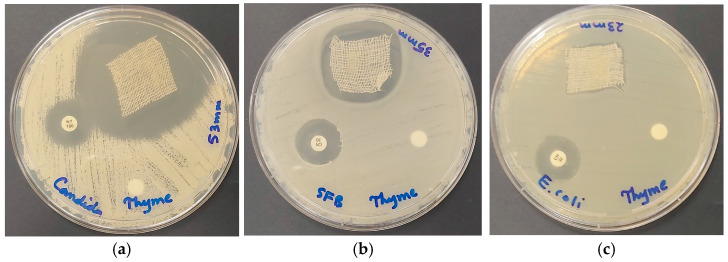
Impregnated sterile bandages with AV-PVP-Thyme-I_2_, with the positive control antibiotics nystatin (100 IU) and gentamicin (30 µg/disc). From left to right: AV-PVP-Thyme-I_2_ (11 µg/mL) against (**a**) *C. albicans* WDCM 00054, (**b**) *B. subtilis* WDCM 00003, and (**c**) *E. coli* WDCM 00013.

**Figure 13 ijms-25-01133-f013:**
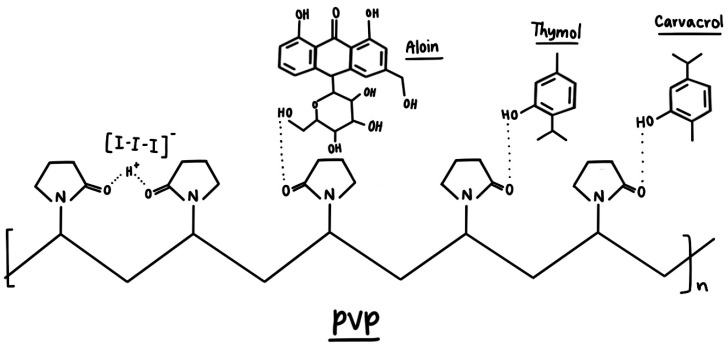
PVP polymeric complex in AV-PVP-Thyme-I_2_.

**Figure 14 ijms-25-01133-f014:**
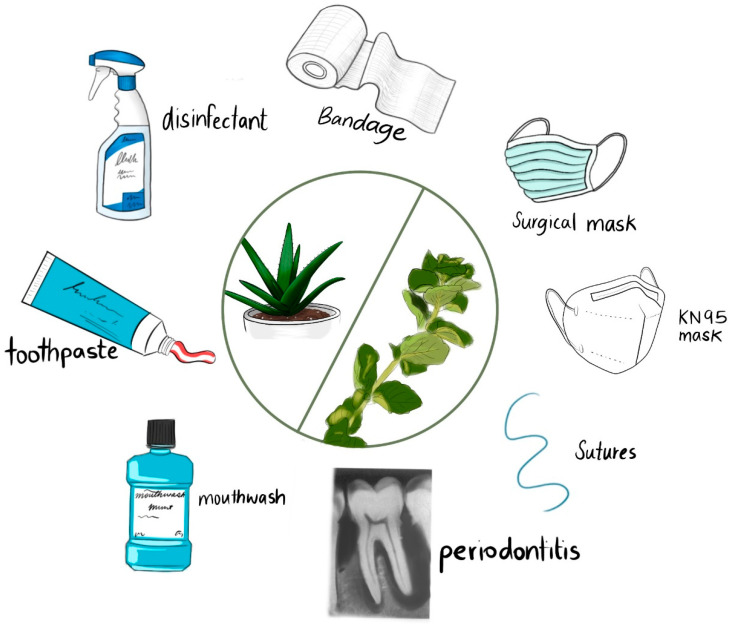
Potential applications of AV-PVP-Thyme-I_2_.

**Figure 15 ijms-25-01133-f015:**
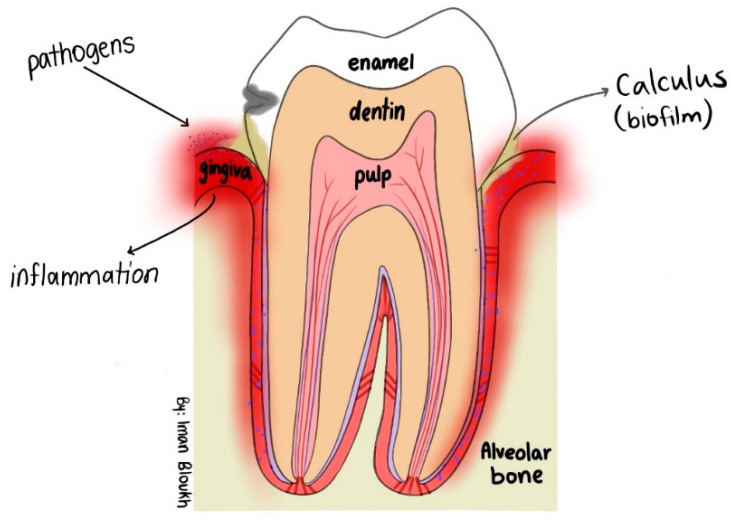
Bacterial colonization in the oral cavity.

**Table 1 ijms-25-01133-t001:** Raman shifts in AV-PVP-Thyme-I_2_ (1) (cm^−1^).

Group	AV-PVP-Thyme-I_2_	[34]	[111]	[118]	[129]
I_2_ [I_2_⋯I^−^]		sh,w 80 *	m 85 *		
		**s169 * ν_as_**	**m 160 * ν_as_**		**s 169ν** ** _s_ **
		w189 * ν			
I_3_^−^		sh,w 61δ_def_	sh 60 δ_def_		
		sh,w 70ν_2bend_	sh,w 75ν_2bend_		
**[I-I-I^−^]**	**s 112ν_1,s_**	**vs 110ν_1,s_**	**s 110ν_1,s_**	**114ν_1,s_**	**vs 111ν_s_**
		vw 222^+^ 2ν_1,s_	vw 221 2ν_1,s_		
I_3_^−^	w 141ν_3,as_				
[I-I⋯I^−^]	w 145ν_3,as_	m 144ν_3,as_	m 144ν_3,as_	144ν_3,as_	m 145ν_s_
		vw 334^+^ν_as_	sh,vw 154ν_3,as_		

ν = vibrational stretching, _s_ = symmetric, _a_ = asymmetric, 1 = stretching mode 1, 3 = stretching mode 3, bend = bending, δ_def_ = deformation. * belong to the same asymmetric, nonlinear unit I_3_^−^ = I_2_⋯I^−^. ^+^ overtones of triiodide ions. vw = very weak, br = broad, s = strong, vs = very strong, m = intermediate, sh = shoulder.

**Table 2 ijms-25-01133-t002:** UV-vis absorption signals in the samples AV-PVP-Thyme-I_2_ (1), AV-PVP-Thyme-I_2_ 18 months old (2), AV-PVP-Thyme (3), PVP-I_2_ (4), pure Thyme macerated for 3 months (5), thymol (6), carvacrol (7), and AV-PVP-Sage-I_2_ [34] (nm).

Group	1	2	3	4	5	6	7	[34]
I_2_	204 vs	206 sh		205 vs				206 vs
I3^−^ as [I-I-I^−^]	289 s,br	289 s,br		290 m,br				290 s,br
I3^−^ as [I-I⋯I^−^]	359 s,br	359 s,br		360 m,br				359 s,br
I_5_^−^				444 w,br				
I^−^	201 sh,vs	201 s		202 sh				202 vs
AV/Aloin	208 vs 230 sh	208 s 230 sh	207 vs 230 sh					206 vs
PVP	208 vs	208 s	204–215 **	207 vs				201–205 **
	209 sh	210 s		210 vs				209 vs
				212 sh				211 br
	**213 s,sh**	**214 sh**	**214 sh**	**215 sh**				215 sh
	217 sh		217 sh	217 sh				
		220 sh		**219 s,sh**				
				221 s,sh				
	224 sh	224 sh	223 sh	224 s				
		227 sh	226 sh					
		233 w		231 sh				
PVP-I_2_	304 w,sh	304 sh	306	304 sh				305 s,sh
Thyme/ Thymol/ Carvacrol/ **Rosmarinic Acid**	202–220 **	202 s	202 sh		202 vs	203 s		
	204 vs	204 s	203 vs		204 s, sh	205 s	203 s	
	**206 s**	**206 s**			**206 s,sh**		204 s	
	207 vs		207 vs		207 vs,sh	207 s		
					209 vs,sh	209 vs	207 vs	
	210 vs,sh					210 vs		
		212 s					210 vs	
			216 sh		214 vs,sh	216 s	212 vs	
			219 sh			220 s	215 vs	
							219 s	
		240 w			241 br,sh		223 m	
	**250**–320 **	**250**–320 **			**250 br,w**			
	277 **	277 **			280 br,vw	277 m		
	**330**–440 **	**330**–440 **			**327 br,vw**		272 m	283 **
	415 br,vw	415 br,vw	415 br,vw		415 br,vw			340 m,sh

V-vis absorption signals with a concentration of 0.11 µg/mL. ** The broad bands overlap, and several peaks related to AV compounds, iodine moieties, and thymol/carvacrol cannot be observed. vw = very weak, br = broad, s = strong, vs = very strong, m = intermediate, sh = shoulder.

**Table 3 ijms-25-01133-t003:** FTIR analysis of AV-PVP-Thyme-I_2_ (A) and PVP-I_2_ (B) in solvent ethanol (cm^−1^).

	ν_1,2_ (O–H)_s,a_ ν (COOH)_a_	ν (C–H)_a_	ν (C-H)_a_	ν (C-H)_s_	ν (C=O)_a_	δ (C-H)_a_ δ (CH_2_) δ (O-H)	ν (C-C)	ν (C-O)	ν (C-O) ν (C-N)
A	3445 vs 3383 vs 3215 vs 3154 s 3148 s 3140 s	2990 vs	2955 vs	2868 vs	1653 m 1753 vw	1456 s δ(CH_3_)_s, in-plane_ 1420 s δ(CH_3_)_a, in-plane_ 881 vs δ(CH_2_)_twisting_ 804 s δ(C-H)_out-of-plane_ 656 s δ(O-H)	1381 s 1326 s	1273 m	1153 sh ν (C-N) 1099 vs ν (C-O) 1036 vs ν (C-O)
B	3350 vs 3154 s 3140 s	2990 vs	2947 vs	2862 vs	1658 m 1753 vw	1456 s δ(CH_3_)_s, in-plane_ 1420 s δ(CH_3_)_a, in-plane_ 880 vs δ(CH_2_)_twisting_ 804 s δ(C-H)_out-of-plane_ 656 s δ(O-H)	1381 s 1331 s	1273 m	1153 sh ν (C-N) 1099 vs ν (C-O) 1038 vs ν (C-O)

ν = vibrational stretching, δ = deformation, s = symmetric, a = asymmetric; absorption intensity: vs = very strong, s = strong, m = medium, vw = very weak, sh = shoulder.

**Table 4 ijms-25-01133-t004:** XRD analysis of the samples AV-PVP-Thyme-I_2_ (1), AV-PVP-Thyme (2), AV-PVP-Sage-I_2_ [34], AV-PVP-I_2_ [35], and in previous reports (2Theta°).

Group	AV-PVP-Thyme	AV-PVP-Thyme-I_2_	[35]	[34]	[120]	[121]	[81]
I_2_	-	-	-	-	25 29 36	24.5 s 25 s 28 s 37 w 38 w 43 w	-
						46 m	
PVP	14.89 vs 24.37 m	14.89 s 24.37 w	10 s 19 s,br	13 s	-	-	-
Thyme	14.92 vs,br 14.98 s,br	14.92 s,br 14.98 m,br	-	**28 s,br**	-	-	11.8 w 15.8 w
	**24.43 w**	**24.43 vw**		42 w,br			**16.6 vs**
	**30.08 s**	**30.08 m**					**18.7 vs**
	**30.14 w**	**30.14 vw**					20.3 m 20.8 m
							24 w **25.4 s**
AV	45.81 w 45.92 vw 49.92 vw 62.54 vw	45.84 vw 45.95 vw 49.95 vw 62.57 vw	14 s 21 s,br 22 s,br	14 s	-	-	38.2 vs 44.4 m 64.9 w

w = weak, br = broad, s = strong, m = intermediate.

**Table 5 ijms-25-01133-t005:** Antimicrobial testing by disc dilution studies of antibiotics (A), AV-PVP-Thyme-I_2_ long-term stability study after 18 months (L1), AV-PVP-Thyme-I_2_ (1,2,3), suture (S), bandage (B), mask blue layer (M^B^), and mask white layer (M^W^). ZOI (mm) against microbial strains by diffusion assay.

Strain	Antibiotic	A	L1 ^+^	1 ^+^	2 ^+^	3 ^+^	S	B	M^B^	M^W^	KN95
*C. albicans* WDCM 00054	NY	16	61	50	35	30 *	3	53	45	55	80
*S. aureus* ATCC 25923	G	28	22	18	17	15	3	32	35	30	38
*B. subtilis* WDCM 00003	G	21	19	15	14	13	3	35	30	40	40
*S. pyogenes* ATCC 19615	G	25	16	13	12	11	2	28	21	25	30
*E. faecalis* ATCC 29212	G	25	15	14	12	11	4	23	23	25	28
*S. pneumoniae* ATCC 49619	G	18	15	11	11	10	1	24	20	26	26
*K. pneumoniae* WDCM 00097	G	30	14	13	12	10	2	25	28	21	24
*E. coli* WDCM 00013	G	23	14	11	10	9	0	23	24	25	20
*P. aeruginosa* WDCM 00026	G	23	8	10	9	8	0	20	16	25	23
*P. mirabilis* ATCC 29906	G	30	0	0	0	0	0	0	0	0	25

^+^ Disc diffusion studies (6 mm disc impregnated with 2 mL of 11 µg/mL (1), 2 mL of 5.5 µg/mL (2), and 2 mL of 2.75 µg/mL (3) of AV-PVP-Thyme-I_2_. A = G (gentamicin) (30 µg/disc). NY (nystatin) (100 IU). Suture (S), bandage (B), and mask (M) were impregnated with 2 mL of 11 µg/mL AV-PVP-Thyme-I_2_. The gray-shaded area represents Gram-negative bacteria. 0 = Resistant. * Further dilution to 1.38 µg/mL yielded a ZOI = 15 mm. There were no statistically significant differences (*p* > 0.05) between row-based values through Pearson correlation.

## Data Availability

Data is contained within the article and Appendix A.

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
