# Peer review of "Green Synthesized Polymeric Iodophors with Thyme as Antimicrobial Agents"

_ijms, 2024, doi:10.3390/ijms25021133_

Round 1

Reviewer 1 Report

Comments and Suggestions for Authors

Antimicrobial interest has been built on tactics in recent years. This is a well-planned research with trials that build on the phenotypic and fundamentals. Overall, I think this work is publishable with few minor comments. I believe this work is publishable with a few refinements.

Line no 29-30: Please reduce the keywords

Line no 55: What is iodine-based formulation's primary purpose?

Line no 755: Have you examined the stability of the molecules that you have developed, namely AV-PVP-Thyme-I2?

Line 787-790: Enhance the amount of preparation you have for FTIR studies. The FTIR was either taken in liquid form or taken in powder form by the author. Mention it

In conclusion section, make it less of a summary and more substantive.

Author Response

Reviewer 1 IJMS 12.1.2023

Antimicrobial interest has been built on tactics in recent years. This is a well-planned research with trials that build on the phenotypic and fundamentals. Overall, I think this work is publishable with few minor comments. I believe this work is publishable with a few refinements.

Dear Reviewer, thank you for your kind words and your valuable comments throughout.

Line no 29-30: Please reduce the keywords

Thank you for your advise. Done. We removed two unnecessary keywords from the list.

Line no 55: What is iodine-based formulation's primary purpose?

Dear Reviewer, thank you for raising this point. I wrote into the manuscript the following sentences to clarify the missing links, which you kindly pinpointed:

“AMR remains a serious threat to the future of mankind because opportunistic pathogens develop resistance against many synthetic antimicrobials in the market. Iodine is a strong microbicide and has until now not been associated with any resistance phenomenon although it is used since history. Therefore, iodine-based formulations have a potential to be used with the purpose to inhibit microorganisms. Adding iodine into the formulation enhances antifungal and antibacterial properties in agreement with our previous works. However, iodine must be protected by PVP in order to have long lasting action. Here, the PVP-I complex acts as slow release reservoir of iodine molecules. Additionally, synergy between plant biocomponents and the PVP-I complex enriches the inhibition portfolio.” (lines 57 pp)

Line no 755: Have you examined the stability of the molecules that you have developed, namely AV-PVP-Thyme-I2?

Yes, dear reviewer, thank you for this comment. It was necessary to mention the long-term study within the materials and methods. Therefore we added:

“The formulation was kept for 18 months under these conditions and characterized in comparison to the fresh sample.” (Lines 771-773)

Line 787-790: Enhance the amount of preparation you have for FTIR studies. The FTIR was either taken in liquid form or taken in powder form by the author. Mention it

Thank you so much, we used liquid sample diluted in ethanol. We added the information:

“The liquid sample was diluted in absolute ethanol and analyzed by FTIR.” (line 800)

In conclusion section, make it less of a summary and more substantive.

Dear Reviewer, we tried not to repeat the discussion in the conclusions. The conclusions usually are to be kept short, but after carefully chaecking, which core aspects are missing, we had to add few lines to clarify again the importance of iodine. Thank you so much for this truly valuable advise:

“As an alternative approach, we add to the mixture of biocomponents molecular iodine. Iodine is a known microbicide and until now, there is no known resistance mechanisms against iodine. Therefore, adding iodine enhances the antimicrobial activities of the plant-biomaterial originating from AV and Thyme. Iodine must be complexed within PVP to prevent its untimely and fast release, mitigate its stability and long-term use. Polymeric complexes of PVP and iodine are already used in many products with cosmeceutical, oral health and pharmaceutical background, as well as excellent antimicrobial agents.” (lines 867-874)

Dear reviewer, we fully appreciate your valuable insight into the topic and thank you very much for highlighting the weak points of our manuscript. Through your advise, we improved the manuscript much more towards a fluent, interesting paper.

Thank you so much

Best regards

Zehra

Reviewer 2 Report

Comments and Suggestions for Authors

The authors performed a on-pot synthesis to prepare AV-PVP-Thyme-I2 with polyvinylpyrrolidone (PVP) complexed Iodine (I2), Thymus Vulgaris L. (Thyme) and Aloe Barbadensis Miller (AV). AV-PVP-Thyme-I2 was measured by SEM/EDS, UV-vis, Raman, FTIR and XRD analysis, which verified the purity, composition and morphology of AV-PVP-Thyme-I2. AV-PVP-Thyme-I2 is strong antifungal- and antibacterial agent against the majority of the tested microorganisms with excellent results on cotton bandages and face masks. The article is interesting. However, some points of the manuscript also should be improved.

1.   AV-PVP-Thyme-I2 should be measured by XPS to verify the valance of important element.

2.   The cytotoxicity should be measured, which is an important factor for the real application.

3.   The component or structure of hymus Vulgaris L. (Thyme) and Aloe Barbadensis Miller (AV) should be offered.

4.   The antimicrobial efficiency of AV-PVP-Thyme-I2 should be compared with other antimicrobial agents.

5.   Please carefully check the manuscript for writing and grammar.

Comments on the Quality of English Language

Minor editing of English language required

Author Response

Reviewer 2 IJMS

The authors performed a on-pot synthesis to prepare AV-PVP-Thyme-I2 with polyvinylpyrrolidone (PVP) complexed Iodine (I2), Thymus Vulgaris L. (Thyme) and Aloe Barbadensis Miller (AV). AV-PVP-Thyme-I2 was measured by SEM/EDS, UV-vis, Raman, FTIR and XRD analysis, which verified the purity, composition and morphology of AV-PVP-Thyme-I2. AV-PVP-Thyme-I2 is strong antifungal- and antibacterial agent against the majority of the tested microorganisms with excellent results on cotton bandages and face masks. The article is interesting. However, some points of the manuscript also should be improved.

Dear Reviewer thank you for your appreciation !

  1. AV-PVP-Thyme-I2 should be measured by XPS to verify the valance of important element.

Dear reviewer, we have unfortunately analytical research lab to get this measurement done in our surroundings. However, we thank you for your advise and will after finding a solution, apply XPS in our next manuscript. Nevertheless, the important element in this study is iodine, which is in elemental form added, changes into triiodides and iodide ions only, as confirmed by the applied analytical methods Raman and UV-vis. These two methods are actually the main and best known methods throughout the literature, which are used to characterize polyiodides in general. Therefore, we did not even have the idea, that XPS is needed. But we can consider it for our further research studies in our lab in the future publications. Thank you so much and we hope, that you may understand our situation.

  1. The cytotoxicity should be measured, which is an important factor for the real application.

Dear Reviewer, again, thank you so much for this advise. Actually, we are since some time trying to get an opportunity to do the cytotoxicity studies for our research publications. Unfortunately, we are not able to do it by ourselves due to lack of a ready infrastructure. Our university is already in the process of setting up and finalizing such a lab. Therefore, we decided to prepare another joint publication with all our previously published formulations to investigate their use and applications in a new paper, once our own lab is ready and functioning. I hope, you understand us. Of course, we would love to prove the application of our formulations and even continue towards in-vivo experimentation as well.

  1. The component or structure of thymus Vulgaris L. (Thyme) and Aloe Barbadensis Miller (AV) should be offered.

Dear reviewer, Thyme consists of different essential oils and polyphenols, from which the most important ones are thymol, carvacrol, rosmarinic acid (lines 70-80). AV contains however more than 75 known ingredients, which play a role as antimicrobial and moisturizing agents. The main ingredients like aloin, acemannan, and mannose, as well as different polyphenols are mentioned in the text (lines 87-99). We investigated the formulation in agreement to our previous publications and found several parallels regarding aloin, rosmarinic acid, thymol and caffeic acid. Still, we noticed, that a new figure illustrating the complexation process could prove supportive for the understanding of our text. A figure sometimes can speak better than 100 sentences. Therefore, we decided to show the complexation of  PVP with few components available in the formulation. Thank you for your  advise. We hope, the manuscript will be more improved by this additional figure.

  1. The antimicrobial efficiency of AV-PVP-Thyme-I2 should be compared with other antimicrobial agents.

Dear reviewer, this is a completely valid and relevant point. Therefore we compared the current formulation AV-PVP-Thyme-I2 with the most similar one available among our previous studies, which is AV-PVP-Sage-I2. Therefore, we added following text:

“We had the same trends for the formulation AV-PVP-Sage-I2 [34]. S. aureus ATCC 25923 shows intermediate inhibition zones of 18, 17 and 15 mm against the title compound and 20, 15, 14 mm for AV-PVP-Sage-I2 (Figure 9b, Table 5) [34]. The Gram-negative K. pneumoniae WDCM 00097 is higher susceptible with ZOI = 13,12 and 10 mm than the other Gram-negative pathogens (Figure 9c, Table 5). The same happens in the formulation AV-PVP-Sage-I2 with 13 and 9 mm [34]. This trend is related to the motility of the pathogens. Accordingly, the most motile, swarming bacteria P. mirabilis ATCC 29906 is not inhibited by AV-PVP-Thyme-I2 in any of the tests except, when impregnated into a KN95 mask (Table 5). The bio-formulation with Sage does not inhibit this motile bacteria as well. The only difference between the formulations AV-PVP-Thyme-I2 and AV-PVP-Sage-I2 is their inhibitory action towards P. aeruginosa WDCM 00026 (Table 5) [34]. The motile bacilli P. aeruginosa WDCM 00026 is resistant towards AV-PVP-Sage-I2 [34]. The title bio-hybrid AV-PVP-Thyme-I2 exerts intermediate inhibitory action against P. aeruginosa WDCM 00026 with ZOI = 11, 10 and 9 mm. Another difference is seen in the susceptibility of B. subtilis WDCM 00003 towards both bio-formulations. The inhibition of B. subtilis WDCM 00003, a rod shaped bacilli, is higher in the title formulation AV-PVP-Thyme-I2 compared to AV-PVP-Sage-I2 with ZOI = 19, 15 and 14 mm against Z = 13, 12 and 11 mm, respectively (Table 5) [34]. In general, the Thyme formulation has higher inhibitory action than the Sage [76-79]. This could be a result of the availability of monoterpenes Thymol and Carvacrol within the title compound AV-PVP-Thyme-I2 [76-79,80-92]. These monoterpenes are known for their antimicrobial actions against many microorganisms [80-92].” (lines 562-583)

  1. Please carefully check the manuscript for writing and grammar.

Done, thank you very much. There were quite many mistakes.

Dear reviewer, thank you for your insightful comments and advises. We are hopeful, that our additional parts improved the manuscript way more. The readers will have a better understanding  due to a more fluent text with interesting points. Regarding the XPS and the cytotoxicity studies, we are planning another manuscript consisting of all our formulations in a comparative study including in-vivo-studies, to present their potential applications as an outcome.

We thank you again for your advises, which improved the manuscript much more.

Best regards

Zehra

Reviewer 3 Report

Comments and Suggestions for Authors

The current manuscript aims to investigate green synthesized polymeric iodophors with thyme as antimicrobial agents. Although the topic is interesting in its scientific field, there are some issues that require the authors’ attention to improve the quality of this particular manuscript before further consideration for publication in a high-quality journal “IJMS”.

Specific comments:

1.         Regarding Figure 2, the authors should also include the control group of PGA without AV-PVP-Thyme-I2 coating in the experimental design for fair scientific comparisons. Otherwise, the audiences are unaware of the necessity of V-PVP-Thyme-I2 in the processing of suture materials.

2.         As stated by the authors, the homogenously coated PGA sutures have potential to be used in surgical operations to prevent surgical site infections. However, the results of material characterization studies cannot justify the aforementioned implication. The authors are encouraged to perform in-depth animal tests to verify their research hypothesis.

3.         Similarly, the authors stated that the formulation of AV-PVP-Thyme-I2 is homogenously distributed on the face mask tissues and does not compromise breathability and air movement (Figures 3a, b). Therefore, their impregnated face mask tissues could be used to mitigate viral load and inflammatory processes in the upper respiratory tract. However, these important issues (i.e., anti-viral and anti-inflammatory activities) should be carefully checked by experimental before drawing a solid conclusive summary. Please improve.

4.         As stated by the authors, the homogenous deposition of AV-PVP-Thyme-I2 on the bandage is a good indicator for the potential use of their impregnated bandages as anti-microbial dressings. However, the results of Figure 4 are irrelevant to the evidences of anti-microbial activity. Please must demonstrate the antimicrobial action of the materials to meet the manuscript title.

5.         As stated by the authors, a mounting number of studies reports successful combinations of plant extracts with different nanoparticles for antimicrobial purposes. In fact, a recent review also targeted at the report on the bioactive nanoparticles as antibacterial agents (DOI: 10.1016/j.cej.2022.134970). If possible, please consider the inclusion of this article in the reference list to support the authors’ claim and balance scientific viewpoint.

6.         Please identify panels a, b, and c in Figure 6.

Author Response

Reviewer 3 IJMS

The current manuscript aims to investigate green synthesized polymeric iodophors with thyme as antimicrobial agents. Although the topic is interesting in its scientific field, there are some issues that require the authors’ attention to improve the quality of this particular manuscript before further consideration for publication in a high-quality journal “IJMS”.

Specific comments:

  1. Regarding Figure 2, the authors should also include the control group of PGA without AV-PVP-Thyme-I2 coating in the experimental design for fair scientific comparisons. Otherwise, the audiences are unaware of the necessity of V-PVP-Thyme-I2 in the processing of suture materials.

Dear Reviewer, thank you for your tip, we added the picture. It is now much better to compare and see, how the impregnation changed the suture.

  1. As stated by the authors, the homogenously coated PGA sutures have potential to be used in surgical operations to prevent surgical site infections. However, the results of material characterization studies cannot justify the aforementioned implication. The authors are encouraged to perform in-depth animal tests to verify their research hypothesis.

Dear Reviewer, this is really true and we agree completely. Therefore we mentioned in different parts of the paper, that additional analytical methods are needed to confirm the application of our formulations. Actually, we are since some time trying to get an opportunity to do the cytotoxicity and animal studies for our research publications. Unfortunately, we are not able to do it by ourselves due to lack of a ready infrastructure. Our university is already in the process of setting up and finalizing such labs currently. Therefore, we decided to prepare another joint publication with all our previously published formulations to investigate their use and applications in a new paper, once our own lab is ready and functioning. I hope, you understand us. Of course, we would love to prove the application of our formulations and even continue towards in-vivo experimentation as well. We hope, you can understand our unique situation.

  1. Similarly, the authors stated that the formulation of AV-PVP-Thyme-I2 is homogenously distributed on the face mask tissues and does not compromise breathability and air movement (Figures 3a, b). Therefore, their impregnated face mask tissues could be used to mitigate viral load and inflammatory processes in the upper respiratory tract. However, these important issues (i.e., anti-viral and anti-inflammatory activities) should be carefully checked by experimental before drawing a solid conclusive summary. Please improve.

Dear reviewer, thank you so much for this very important issue. Actually, we discussed the results of our 10 reference strains in the text, but did not clearly mention the need for more investigations. The antimicrobial activities against our 10 reference strains were proven, therefore, we can only claim antifungal and antibacterial action. Actually, in order to study the anti-viral capabilities, we must perform tests. Therefore, we added in the needed parts new parts. Thank you for this important issue.

“However, further studies are needed to confirm antiviral activity of our formulation and will be part of our future investigations.” (lines 227-229)

We added following “Antiviral tests were not performed and are planned to be investigated in future studies in our group.” (lines 657-658)

“Future investigations should include cytotoxicity-, cell culture-, in-vivo- and antiviral studies to verify the suitability of mentioned applications.” (lines 748-750)

  1. As stated by the authors, the homogenous deposition of AV-PVP-Thyme-I2 on the bandage is a good indicator for the potential use of their impregnated bandages as anti-microbial dressings. However, the results of Figure 4 are irrelevant to the evidences of anti-microbial activity. Please must demonstrate the antimicrobial action of the materials to meet the manuscript title.

Dear Reviewer, thank you for your valuable comment. Yes, true, the material is very suitable for bandages. We impregnated bandages with the formulation and performed the tests. The antimicrobial properties are shown on Table 5 and Figure 12. The results are very interesting and imply a possible use in bandages.

  1. As stated by the authors, a mounting number of studies reports successful combinations of plant extracts with different nanoparticles for antimicrobial purposes. In fact, a recent review also targeted at the report on the bioactive nanoparticles as antibacterial agents (DOI: 10.1016/j.cej.2022.134970). If possible, please consider the inclusion of this article in the reference list to support the authors’ claim and balance scientific viewpoint.

Dear Reviewer, thank you so much for the excellent review. We included it as [23].

  1. Please identify panels a, b, and c in Figure 6.

Thank you so much ! We did not notice this mistake. Of course it is now done.

Dear Reviewer, thank you so much for your insight and valuable comments, which helped to improve our manuscript much more.

Best regards

Zehra

Round 2

Reviewer 3 Report

Comments and Suggestions for Authors

In my opinion, the revised manuscript is suitable for publication in “IJMS”.